# Roles of HIF and 2-Oxoglutarate-Dependent Dioxygenases in Controlling Gene Expression in Hypoxia

**DOI:** 10.3390/cancers13020350

**Published:** 2021-01-19

**Authors:** Julianty Frost, Mark Frost, Michael Batie, Hao Jiang, Sonia Rocha

**Affiliations:** 1Department of Molecular Physiology and Cell Signalling, Institute of Systems, Molecular and Integrative Biology, University of Liverpool, Liverpool L69 7ZB, UK; Julianty.Frost@liverpool.ac.uk (J.F.); Mark.Frost@liverpool.ac.uk (M.F.); M.Batie@liverpool.ac.uk (M.B.); 2Centre for Gene Regulation and Expression, School of Life Sciences, University of Dundee, Dundee DD1 5EH, UK; h.y.jiang@dundee.ac.uk

**Keywords:** hypoxia, 2-OG dioxygenases, chromatin, transcription, translation, cancer

## Abstract

**Simple Summary:**

Hypoxia—reduction in oxygen availability—plays key roles in both physiological and pathological processes. Given the importance of oxygen for cell and organism viability, mechanisms to sense and respond to hypoxia are in place. A variety of enzymes utilise molecular oxygen, but of particular importance to oxygen sensing are the 2-oxoglutarate (2-OG) dependent dioxygenases (2-OGDs). Of these, Prolyl-hydroxylases have long been recognised to control the levels and function of Hypoxia Inducible Factor (HIF), a master transcriptional regulator in hypoxia, via their hydroxylase activity. However, recent studies are revealing that such dioxygenases are involved in almost all aspects of gene regulation, including chromatin organisation, transcription and translation.

**Abstract:**

Hypoxia—reduction in oxygen availability—plays key roles in both physiological and pathological processes. Given the importance of oxygen for cell and organism viability, mechanisms to sense and respond to hypoxia are in place. A variety of enzymes utilise molecular oxygen, but of particular importance to oxygen sensing are the 2-oxoglutarate (2-OG) dependent dioxygenases (2-OGDs). Of these, Prolyl-hydroxylases have long been recognised to control the levels and function of Hypoxia Inducible Factor (HIF), a master transcriptional regulator in hypoxia, via their hydroxylase activity. However, recent studies are revealing that dioxygenases are involved in almost all aspects of gene regulation, including chromatin organisation, transcription and translation. We highlight the relevance of HIF and 2-OGDs in the control of gene expression in response to hypoxia and their relevance to human biology and health.

## 1. Introduction

The importance of oxygen for energy production in multicellular organisms has been appreciated since the identification of the mechanism of oxidative phosphorylation located in the mitochondria. Reduction in oxygen availability, or hypoxia, is therefore either a danger signal or a cue for physiological processes such as development. Every cell has a different threshold for mounting a hypoxia response, although the exact mechanisms dictating this threshold are currently not well understood. Given the importance of oxygen, cells have evolved sophisticated mechanisms to sense and respond to hypoxia, in order to minimise damage, preserve energy and, when possible, adapt to the new oxygen supply normality.

The main transcription factor activated under low oxygen conditions, called Hypoxia Inducible Factor (HIF), was identified in 1992 [1]. HIF is composed of a heterodimer of HIF-α (of which there are three isoforms encoded by three different genes, *HIF-1α*, *HIF-2α* and *HIF3-α*) and HIF-1β [2]. HIF3-α, unlike the other HIF-αs, lacks a transactivation domain, and is thought to act either as a dominant-negative or transcription-independent regulator of the hypoxic response [3]. HIFs control many genes, most of which are crucial for cell survival and adaptation to low oxygen conditions [2]. Under pathological conditions, such as cancer and altitude sickness, induction of some of these genes by the HIF transcription factors has been linked to disease progression and treatment resistance [4]. In addition, HIF can also be induced by non-oxygen dependent mechanisms, such as inflammation [5]. This is particularly relevant for human cancers, where hypoxia and inflammation often co-occur [6].

The detailed mechanism leading to the activation of HIF was unravelled in 2001 [7,8], building on accumulated knowledge of the role of Von Hippel–Lindau Tumor Suppressor (VHL) in this process [9,10]. HIF-α, under normal oxygen conditions, is continually transcribed and translated, but rapidly degraded by the ubiquitin dependent proteasomal system (Figure 1). Ubiquitination is promoted by the E3-ligase composed of VHL, Ring-Box 1 (RBX1), Cullin 2 (CUL2) and Elongin B/C (ELOCB/C) [4]. VHL affinity toward HIF-α is dramatically increased by the presence of a specific post-translational modification, more precisely two separate proline hydroxylation events (Figure 1), mediated by Prolyl-Hydroxylases (PHDs). PHDs are part of the 2-oxoglutarate (2-OG) dependent dioxygenase (2-OGD) superfamily of enzymes, requiring oxygen, iron (II) and 2-OG for activity [11]. Mammals possess three PHDs, PHD2 (gene name *EGLN1*), PHD3 (gene name *EGLN3*) and PHD1 (gene name *EGLN2*) [7].

Biochemical characterisation revealed that PHDs have low affinity for molecular oxygen. Low affinity for molecular oxygen signifies that when oxygen availability is reduced, these enzymes are quickly inhibited, leading to HIF stabilisation and activation of target genes. PHD inhibition in hypoxia has resulted in them being termed molecular oxygen sensors in the cell [12]. Additionally, many more 2-OGDs have been identified, most of which act independently of HIF but play a role in coordinating the cellular response to hypoxia. These include Factor Inhibiting HIF1 (FIH), Jumonji C (JmjC)-domain containing demethylases (JmjC demethylases) (which demethylate both histones and non-histone proteins), Ten-Eleven Translocation (TET) enzymes (mediators of DNA-demethylation), and RNA-demethylases (reported in vitro for 2-ODGs are shown is Table 1 and Appendix A). Given the function of some of these enzymes, it is conceivable that hypoxia could influence all aspects of gene expression, from chromatin structure and epigenetics to RNA biology, translation and protein turnover (Figure 2). This perfectly equips the cell when faced with hypoxia. Under such conditions, the cell must make a coordinated effort to allow for restoration of oxygen homeostasis, while reducing energy expenditure if it is to survive.

Genetic models in several model organisms have helped identify the key roles of HIFs as well as 2-OGDs in development and disease (Table 2; Appendix A). Furthermore, genome wide mapping techniques such as chromatin immunoprecipitation followed by sequencing (ChIP-seq), RNA sequencing (RNA-seq), and Chromatin capture have more recently been used to better understand how cells responds to changes in oxygen, but also in response to 2-OGD inhibition ([13,14] and reviewed in [15,16]).

In this review, we highlight the importance of oxygen sensing in coordinating an efficient response to hypoxia. We discuss the relevance of HIF transcription factors, and roles of 2-OGDs in controlling almost all aspects of gene expression from chromatin structure, to transcription, translation and post-translational modifications.

## 2. Effects of Hypoxia on Gene Transcription

It is now appreciated that the cellular and organism response to hypoxia involves profound changes to gene expression, with vast changes in gene transcription being detected in all systems studied.

### Hypoxia-Induced Changes to Transcription Are Largely Mediated by HIF

As mentioned earlier HIFs are the master regulators of gene transcriptional changes in hypoxia and this has been extensively reviewed previously [2,11,203]. The discovery of hypoxia responsive genes and HIF targets has been driven greatly by transcript profiling and genome occupancy technologies, including microarrays, RNA-seq and ChIP-seq (reviewed in [15,16,204]). These analyses of publicly available transcriptional datasets [205,206] have shed light on cell type differences in hypoxia responsive genes and HIF targets on a genome wide scale, as well as identifying hypoxic signatures conserved across multiple cell types. Why different cell types have different transcriptional responses to hypoxia and HIF target genes is an important question for the field, with chromatin structure organisation thought to be a major factor in conferring specificity. Further to HIF isoform expression and activity, evidence points towards pre-established chromatin accessibility and local chromatin environment, including RNA pol II availability, pre-existing promoter enhancer interactions at HREs, and HRE DNA methylation status, as cell-type specificity determinants of hypoxia transcriptional responses (reviewed in [15,16,204,207]).

Whilst most HIF binding sites are at proximal promoters, binding to distal intergenic regions also occurs and HIF can regulate transcription of long genomic intervals, interacting at promoter-enhancer loops [208,209,210]. Although there is no doubt HIF is the main transcription factor controlling hypoxia-induced transcriptional changes, there is involvement of other transcription factors, including Nuclear Factor-κB (NF-κB), Tumor Protein p53 (p53), MYC Proto-Oncogene (MYC) and Activator Protein 1 (AP1), which function in the regulation of the hypoxia response via HIF-dependent and independent pathways (reviewed in [211]). In addition, HIF mostly acts as an activator of transcription, and thus most of the observed hypoxia-induced gene silencing is either independent of HIF, or via indirect mechanisms, including through the actions of chromatin remodeller complexes, co-repressor complexes or induction of other transcription factors (reviewed in [212]). HIF induced genes are involved in a variety of different pathways involved in restoration of oxygen homeostasis [2,11,203]. Importantly, many 2-OGDs, including *PHD2* [213] and *PHD3* [214], several JmjC demethylases (reviewed in [215]), and *TET1* [216] and *TET3* [216,217], are HIF target genes that have been shown to be transcriptionally upregulated in response to hypoxia.

Although we now have access to a vast number of transcriptomic studies of cells in hypoxia, very few studies have investigated the proteomic changes under such an important stress, using techniques ranging from the “old-fashioned” two-dimensional electrophoresis (2-DE) coupled with MALDI-TOF-TOF-MS [218,219,220], to the more accurate and robust quantitative multiplexed proteomics workflow [221,222,223,224], which usually combines isobaric labelling, two-dimensional liquid chromatography (2D-LC) and high resolution MS. The few studies available have shown that hypoxia exposure of different cells and tissues possesses broad effects at the whole proteome level, including changes to the protein expression of annexin family [218], glycolytic and antioxidant enzymes [219,220], transcription factors [225], heat shock proteins, S100 family proteins [221], and also other proteins involved in the TCA-cycle [226], metabolism [227] and immune response [222]. A proteomic study has revealed novel non-HIF hypoxia regulators, including the chromatin organizer protein Heterochromatin Protein 1 Binding Protein 3 (HP1BP3), which mediates chromatin condensation [223]. The use of multi-omics techniques (transcriptomics, proteomics and metabolomics) and analysis using integrated bioinformatics identified consistent changes to proteins and metabolites in heart tissues under antenatal hypoxia. These proteins and metabolites are involved in energy metabolism, oxidative stress and inflammation-related pathways, required for the reprogramming of the mitochondrion [224].

## 3. Chromatin Regulation in Hypoxia

Central to the hypoxia response is the activation of a dynamic transcriptional programme. HIF transcription factors are the primary mediators of hypoxia induced gene transcriptional changes (reviewed in [11]). Further to HIF stabilisation and activation under low oxygen tensions, the chromatin landscape also plays a complex role in coordinating hypoxia inducible changes to gene transcription. Most aspects of chromatin regulation are altered in response to low oxygen, including histone methylation and acetylation, DNA methylation, actions of chromatin remodeller complexes and non coding RNAs, histone eviction and incorporation of histone variants, and chromatin accessibility (reviewed in [15,16,203]). However, this is still a vastly unexplored aspect of the hypoxia response. As mentioned above, in addition to PHDs, TETs (mediators of DNA demethylation), and JmjC demethylases (histone and non-histone protein demethylases), are also 2-OGDs. Recent studies demonstrate the potential of TETs and JmjC demethylases to function as molecular oxygen sensors, directly linking oxygen sensing to transcriptional control via epigenetics in cells. Below we summarise DNA and histone methylation changes in hypoxia, with a focus on oxygen sensing mechanisms via TETs and JmjC demethylases. (Figure 3).

### 3.1. JmjC Demethylases and Chromatin Regulation in Hypoxia

Histone methylation is a recognised mechanism controlling chromatin structure and is associated with regulation of gene transcription, with some marks clearly leading to open chromatin, while others are firmly associated with closed conformation [228]. The family of enzymes predominantly responsible for histone demethylation are JmjC demethylases, which are part of the JmjC-domain containing group of 2-OGDs that includes demethylases and hydroxylases (JmjC 2-ODGs)). In vitro studies demonstrate the varied oxygen sensitivities of these enzymes, some are potentially direct molecular oxygen sensors (Table 1 and Appendix A). Oxygen affinities, oxygen availability and protein expression levels will likely dictate JmjC demethylase activities in hypoxia. Several groups have reported increases in total levels of histone methylation modifications in response to hypoxia across a range of human and mouse cell types and human tumours using immunoblotting, immunohistochemistry, immunofluorescence and quantitative proteomics ([229], reviewed in [16]). ChIP-sequencing approaches have revealed site-specific, hypoxia-induced changes in Histone (H)3 Lysine (K)4 trimethylation (me3) [230,231], H3K36me3 [230] and H3K27me3 [25,231], which correlate with changes in gene expression. There is now evidence, through the use of in vitro and in cell histone demethylation assays, in coordination with mutagenesis analysis, gene expression analysis and histone methylation analysis, that Lysine Demethylase (KDM) 6A (KDM6A) [25] and potentially KDM5A [230], which demethylate H3K27me3 and H3K4me3 respectively, are inhibited by reduced oxygen levels in hypoxia. These result in increased histone methylation modifications, which coordinate hypoxia inducible gene transcriptional changes (Figure 3) and hypoxia induced cellular responses. Specifically, KDM6A inhibition in hypoxia triggers hypermethylation of H3K27me3 at a subset of hypoxia repressed gene promoters, reducing their expression. Conversely, potential KDM5A inhibition in hypoxia, triggers H3K4me3 hypermethylation at a subset of hypoxia inducible gene promoters, this precedes increases in their expression in hypoxia and is required for their full transcriptional activation in hypoxia. In cell and in vitro H3K9me3 demethylation assays have also revealed that the demethylase activity of KDM4A is highly sensitive to oxygen concentrations over physiologically relevant ranges [24]. Although there are a range of reported in vitro oxygen affinities, (Appendix A) the data from cells show that KDM4A activity is sensitive to oxygen levels. There is evidence that reduced histone demethylase activity of KDM4A in severe hypoxia (<0.5% oxygen), along with KDM4C, triggers increased expression of a subset of androgen receptor target genes [232]. Thus, KDM4A can also be classed as an oxygen sensor. Interestingly, KDM4A has been shown to positively regulate HIF-1α levels via H3K9me3 demethylation at the *HIF1A* gene locus, this effect is observed in mild hypoxia (2% oxygen), but impaired at severe hypoxia (<0.1% oxygen) [233]. This may provide a mechanism of maintaining HIF-1α levels in conditions of mild hypoxia specifically, where there is still residual PHD activity. Future work should investigate if the oxygen sensitive H3K9me3 demethylase activity of KDM4A is linked to control of gene expression and chromatin regulation in hypoxia. Importantly, some JmjC demethylases remain active at low oxygen concentrations and function in hypoxia through their histone demethylase activity. KDM4C [234] and KDM3A [235,236] display HIF coactivator activity in hypoxia via demethylation of H3K9 at HIF target gene promoters, facilitating transcriptional activation at these genes. The co-activator role of KDM3A in hypoxia, via retained histone demethylation activity appears to be context dependent, as other groups have reported reduced activity KDM3A in hypoxia [237]. Furthermore, many JmjC histone demethylases are HIF target genes that are upregulated in hypoxia (reviewed in [215]), this is thought in part to be a compensatory mechanism to counteract reduced demethylase activity, acting as a hypoxia feedback loop similar to what is seen with transcriptional upregulation of PHD2/3 by HIF. Thus, there is complex crosstalk between histone methylation, gene expression and hypoxia, mediated in part by JmjC demethylases. However, further characterisation of the oxygen sensitives of JmjC demethylases is needed.

### 3.2. TET Mediated DNA Demethylation Functions in Hypoxia

TETs, of which there are three variants—Ten Eleven Ten translocation Methylcytosine Dioxygenase 1 (TET1), TET2 and TET3—are 2-ODGs that function as hydroxylases, mediating mammalian DNA demethylation through catalysing the oxidation of 5-methylcytosine (5mc) to 5-hydroxymethylcytosine (5hmC), 5-formylcytosine (5fC) and 5-carboxylcytosine (5caC). These TET oxidised derivatives of 5mc can be demethylated by mechanisms of active and passive demethylation (reviewed in [238]). DNA methylation can repress gene transcription, consequently tumour hypoxia results in aberrant DNA methylation profiles promoting tumour suppressor gene silencing (hypermethylation) [239] and oncogene activation (hypomethylation) [240]. Recently, using in vitro biochemical binding assays, in vivo studies on HIF binding and DNA methylation status in human cancer cell lines, and in silico structural modelling, D’Anna and colleagues find that DNA methylation at HREs impairs HIF binding, and HRE DNA methylation status is a key factor in determining cell type specific transcriptional responses to hypoxia [241]. There is heterogeneity regarding the effects of hypoxia, both in cell models and in tumours, on global DNA methylation levels and TET activity (reviewed in [203,242]). As such, whether TETs are impaired or functionally active in hypoxia, and the consequences this has for gene transcription appear highly context dependent. Researchers have shown TET activity in hypoxia, and HIF-dependent TET upregulation and coactivator functions have been demonstrated at hypoxia-inducible genes (reviewed in [203,242]). TET activity in low oxygen environments is supported by in vitro oxygen affinities of TET1 and TET2 (Table 1 and Appendix A), as well as the known roles of TETs in the bone marrow and during development where oxygen tensions are low [243,244]. Conversely, Thienpont et al. showed that severe hypoxia (0.5% oxygen) in human and murine cells and tumour hypoxia in multiple human tumours causes DNA hypermethylation at gene promoters correlating with gene silencing at a subset of hypoxia repressed genes and gene silencing linked to hypoxia associated tumour progression. DNA hypermethylation was attributed to oxygen dependent reduction in TET1 and TET2 activity in hypoxia, with a 50% reduction in activity observed at 0.3% oxygen for TET1 and 0.5% for TET2 in vitro. Thus, TET1 and TET2 may be characterised as tumour oxygen sensors, and depending on the context of oxygen deprivation, may remain active in hypoxia environments or display inhibition. However, more work is needed to establish the oxygen dependence of TET activity in cells and in vivo and the physiological contexts in which TETs can sense changes in oxygen availability, as well as the consequences this has for DNA methylation, gene transcription and cellular responses. Indeed, the seemingly contradictory roles for TETs in hypoxia from studies to date may be dependent on the different cell models used and timing/severity of hypoxic stimulus.

While there is growing evidence for a dynamic role of chromatin/epigenetics in sensing and responding to hypoxia to facilitate transcriptional changes, via HIF-dependent and -independent mechanisms, efforts to elucidate molecular mechanisms underpinning such changes and the extent to which chromatin/epigenetic changes are required for coordinating hypoxia/HIF transcriptional effects are ongoing. The discoveries of oxygen sensing by TETs and JmjC demethylases provide an exciting link between oxygen availability and chromatin regulation, and future work on oxygen sensing by chromatin will be essential for better understanding hypoxia driven processes.

## 4. Effects of Hypoxia on Protein Levels

Protein levels and function are key aspects to achieve the appropriate hypoxia response. Although transcription is important, mechanisms controlling protein levels and function supersede any changes in transcriptional output. In hypoxia, mechanisms exist that control translation, but also post-translation aspects of protein function.

### 4.1. Translation Is Globally Repressed in Response to Hypoxia

In addition to the regulation of gene transcription in hypoxia, gene expression is also controlled through regulation of translation (Figure 4). The cellular response to hypoxia includes a reduction in the energy expenditure of the cell due to limited ATP production through oxidative phosphorylation. This adaptation results in a reversible global decrease in energy-expensive protein synthesis (reviewed in [2]). This inhibition of translation is a highly regulated response to low oxygen levels preceding ATP depletion (reviewed in [2]).

Global inhibition of protein expression is largely regulated at the point of translation initiation through two pathways. Firstly, Mechanistic Target of Rapamycin Kinase (mTOR) is inhibited by DNA Damage Inducible Transcript 4 (DDIT4) (a HIF target gene). DDIT4-dependent release of TSC Complex Subunit 2 (TSC2) from 14-3-3 binding proteins leads to mTOR inhibition. This allows the formation of an active TSC1-TSC2 dimer that inhibits the phosphorylation of Ribosomal Protein S6 Kinase B1 (RPS6KB1) by the mTOR protein complex, which in turn inhibits the phosphorylation of Ribosomal Protein S6 (RPS6), part of the 40S ribosomal subunit required for initiation [245]. mTOR inhibition also causes hypophosphorylation of Eukaryotic Translation Initiation Factor (eIF) 4E Binding Protein 1 (eIF4EBP1), allowing the sequestering of eIF4E and decrease of 5’cap-dependent initiation [246,247]. Secondly, PKR-like Endoplasmic Reticulum Kinase (PERK) (gene name *EIF2AK3*) is phosphorylated and activated, which subsequently phosphorylates eIF2α at S51, causing effective inactivation [248]. Phospho-eIF2α prevents binding with eIF2B for exchange of GDP for GTP, therefore remaining in an inactive state and preventing subsequent rounds of translation from the mRNA [249]. eIF2α normally recruits the initiator aminoacylated tRNA to the 40S ribosome, thus limiting global initiation of translation. This second mechanism is independent of HIF and as of yet has not been linked to any 2-OGD.

Inhibition of translation is also regulated at the stage of polypeptide elongation. Elongation is inhibited by phosphorylation of Eukaryotic Elongation Factor 2 (eEF2) at T56 by eEF2 Kinase (eEF2K) [250]. This process has been shown to be dependent on mTOR and 5′-AMP-activated protein kinase catalytic subunit alpha-1 (AMPK) (gene name PRKAA1) [251,252]. Interestingly, eEF2 kinase (eEF2K) is also regulated by hydroxylation by PHD2 at P98 in an oxygen-dependent manner [253]. In hypoxia, when PHD2 inhibited, eEF2K activity is induced.

In addition to PHD2 dependent hydroxylation, there are several other hydroxylation reactions involved in the regulation of translation, catalysed by other 2-OGDs. Hydroxylation is important for the biosynthesis of tRNAPhe with position 37 requiring a hypermodified nucleoside Wybutosine (yW) that can be hydroxylated to form hydroxywybutosine (OHyW) by the JmjC hydroxylase, TRNA-YW Synthesizing Protein 5 (TYW5), which maintains translational fidelity. It is currently unknown whether TYW5 is responsive to the level of oxygen, but its transcription is decreased in hypoxia, [254] linking hypoxia to a decreased accuracy of translation, which globally decreases the successful translation of proteins [255].

The rate and accuracy of translation are positively regulated by hydroxylation of the central translation machinery. JmjC hydroxylases Ribosomal Oxygenase 2 (RIOX2) and RIOX1 hydroxylate histidyl residues in ribosomal proteins, with RIOX2 and RIOX1 hydroxylating Ribosomal Protein L (RPL) 27a (RPL27A) and (RPL8), respectively. The hydroxylation occurs at residues close to the peptidyl transfer centre, thereby increasing translation efficiency [256]. *RIOX1* and *RIOX2* transcription is reduced in hypoxia [254,256]. Furthermore, RPL8 hydroxylation is also reduced in hypoxia [256]. However, it is not yet known whether these enzymes are inhibited by low oxygen levels, or lower hydroxylation is solely due to lower transcription. Additionally, hydroxylation of 40S Ribosomal Protein S23 (RPS23) by the 2-OGD, 2-Oxoglutarate and Iron Dependent Oxygenase Domain Containing 1 (OGFOD1), is required for efficient translation [257]. OGFOD1 transcription is also decreased in hypoxia, but the enzyme remains mostly active, even in acute hypoxia [258], suggesting this mechanism is not through direct 2-OGD oxygen sensing.

Efficient decoding of the mRNA during translation requires the JmjC hydroxylase, AlkB Homolog 8, TRNA Methyltransferase, ALKBH8, which hydroxylates tRNA at the wobble position [259,260]. This 2-OGD has yet to be linked to hypoxia, though it would be interesting to investigate its oxygen sensitivity. Finally, lysyl hydroxylation of eukaryotic release factor 1 (eRF1) by the JmjC hydroxylase Jumonji Domain Containing 4 (JMJD4) is required for proper termination of translation [261], although its activity is not significantly inhibited in hypoxia.

### 4.2. Utilising Proteomics for the Identification of Non-Histone Protein PTMs

Proteomics approaches have revealed hypoxia induces changes to many post translational modifications (PTMs) on non-histone proteins, such as proline hydroxylation [262,263] (regulating protein levels and interactions), phosphorylation [264,265], SUMOylation [266], acetylation [267], glycosylation [268], nitration [269] and nitrosylation [270], all of which regulate protein functions in different ways.

As one of the most widely studied PTMs, phosphorylation of some transcriptional factors and regulators has been found to be changed under various hypoxic conditions, including CAMP Responsive Element Binding Protein 1 (CREB1), NFKB Inhibitor Alpha (NFKBIA), a regulator of NF-κB, and HIF (reviewed in [271]). More recently, through the analysis of phospho-proteomics in renal clear cell carcinoma cells under a VHL-independent hypoxic response, up-regulation of known biomarkers of RCC and signalling adaptor were found. Meanwhile, such hypoxic responses decreased the phosphorylation on intracellular Carbonic Anhydrase 2 (CA2), which might be an unusual way to control the CA2 expression and enhance the activity of the NF-κB pathway, resulting in loss of VHL [264].

In recent years, non-HIF targets have been identified to be hydroxylated on prolines by PHDs (reviewed in [15]), resulting in their degradation and/or changes to downstream activity including Centrosomal Protein 192 (CEP192) [272], Forkhead Box O3 (FOXO3) [273] and Adenylosuccinate Lyase (ADSL) [274] by PHD1, Actin Beta (ACTB) by PHD3 [275], and AKT Serine/Threonine Kinase 1 (AKT1) [93], TANK Binding Kinase 1 (TBK1) [276] and Scm-like with Four Malignant Brain Tumour Domains 1 (SFMBT1) [277] by PHD2 [278]. Despite these exciting studies, a recent study investigating around 20 of the reported non-HIF substrates, failed to detect any hydroxylation in vitro [279]. This adds to the complexity of identifying novel PHD substrates. Interestingly, a new study indicated that prolyl-hydroxylation could be crucial for GMGC kinase activation [280]. This could imply an intricate interplay between these two types of PTMs, suggesting yet another role for oxygen-dependent signalling in the cell. Thus, unbiased proteomic studies on novel PTMs sites [262,281], system-wide analysis of PHDs substrates other than HIF-α [263] and crosstalk of PTMs on PHDs targets in response to hypoxia are now emerging.

## 5. Other Potential Roles of 2-OGDs in the Hypoxia Response

Further to known and potential oxygen sensing roles of JmjC 2-ODGs (JmjC demethylases and hydroxylases) in the regulation of chromatin and translation discussed earlier, there are other functions of JmjC 2-ODGs that may influence the hypoxia response (Figure 5). One of the most prominent of such enzymes is the JmjC 2-OGD, JMJD6, Arginine Demethylase and Lysine Hydroxylase, which has unique activities as both an arginine demethylase (acting on mono- and symmetric/asymmetric di-methylated arginines) and lysine hydroxylase [282,283]. JMJD6 expression is increased in hypoxic conditions in the placenta and can downregulate HIF-1α [284], though it has been found to operate in diverse pathways. JMJD6 can promote the formation of stress granules through demethylation of Arg-435, Arg-447 and Arg-460 of G3BP Stress Granule Assembly Factor 1 G3BP1, and subsequent de-repression of the protein, resulting in the cytoplasmic sequestering of stalled mRNA-ribosome complexes to reversibly prevent mRNA degradation [285,286]. This would allow a fast re-start of protein synthesis when oxygen homeostasis is restored. JMJD6 also regulates mRNA splicing through hydroxylating the splicing regulatory (SR) proteins LUC7 Like 2, Pre-MRNA Splicing Factor (LUC7L2) on Lys-269, U2 Small Nuclear RNA Auxiliary Factor 65 (U2AF65) on Lys-15 and Lys-276 [287], and multiple fragments of Serine and Arginine Rich Splicing Factor 11 (SRSF11) between residue 271 to 372 [283]. The SR proteins are involved in exon definition and alternative splicing, with SRSF11 hydroxylation resulting in skipping of most 5’ exon, and hydroxylation of U2AF65 possibly enacting pre-mRNA looping in order to present to the splicing machinery different *cis* splice enhancer or silencer sequences [288]. However, this only occurs for selected mRNAs and is not a global effect [288]. Nevertheless, this mechanism would allow selection of alternate splice variants as a response to hypoxia. JMJD6 can also interact with both Bromodomain Containing 4 (BRD4) and the positive Transcription Factor Elongation Factor b (P-TEFb) complex [289], eventually resulting in the release of paused DNA polymerase II and resumption of mRNA synthesis at specifically regulated genes [288]. This implies that hypoxia could use this mechanism to stall transcription of genes that are not required for the stress response to hypoxia and would allow a re-start of gene expression when oxygen levels are restored. JMJD6 hydroxylates p53 on Lys382, repressing its transcriptional activity [290]. JMJD6 also auto-hydroxylates itself on the N-terminus and this is required for homo-oligomerisation [291]. The function of the homomultimer of JMJD6 is unknown, but speculated to play a role as a scaffolding protein in the nucleus [292].

Another JmjC hydroxylase, KDM8, can hydroxylate arginine residues in both RCC1 Domain Containing 1 (RCCD1) and RPS6 [293]. KDM8 also has amine- and endo-peptidase activity with specificity towards histone H3 monomethyl-lysine residues, especially of the histone variant H3F3A [294], and arginine methylated histone tails [295,296]. It had previously been reported that KDM8 can demethylate H3K36me2 [297], though considering its arginine demethylation is achieved through trimming of the histone tail, this may be the mechanism of observed decreases in lysine methylation. Although not necessarily dependent on its hydroxylation activity, KDM8 is required for cell proliferation and chromosomal stability [298], and can negatively regulate p53, affecting gene expression and control cell cycle and proliferation [299,300]. Recently, a biochemical function has been assigned to JMJD7 as a lysyl hydroxylase, which targets Developmentally Regulated GTP Binding Protein 1 (DRG1) and DRG2 that are part of the Translation Factor (TRAFAC) family of GTPases, and could affect their binding with messenger, or ribosomal RNA, though this requires further investigation [301]. Similar to KDM8, JMJD7 also possesses endo- and amine-peptidase activity, preferentially cleaving mono- and di-methylated arginine residues of H2, H3, and H4 [295,296].

Several JmjC demethylases have been found to have additional, non-histone substrates. For example, KDM2A represses NF-κB activity via demethylation of RELA, providing a possible link to hypoxia and inflammation crosstalk. [302]. KDM3A was recently found to function as a molecular oxygen sensor, with its demethylation of K224me1 on peroxisome proliferator-activated receptor gamma coactivator-1 alpha (PGC-1α) (which facilitates its activity) impaired under low oxygen conditions [237]. PGC-1α is a transcriptional coactivator promoting mitochondrial biogenesis and oxidative metabolism, the aforementioned work neatly links mitochondrial function to the hypoxia response via the oxygen sensing function of KDM3A controlling PGC-1α activity, further highlighting the importance of JmjC 2-OGDs in the hypoxia response. KDM3A can also demethylate K372me1 of p53, which decreases its association with chromatin and represses its proapoptotic activity [303]. KDM3B, in addition to its substrate H3K9me2, can also demethylate mono- or di-methylated arginine residues, H4R3me2 [127]. It is more than likely that other JmjC demethylases interact and directly demethylate additional transcription factors which may coordinate transcriptional responses to hypoxia. However, unbiased analysis is required to fully assess this aspect of hypoxia-induced gene regulation. The KDM4 family of demethylases appears to have higher activity against trimethylated non-histone targets than its canonical substrates, H3K9me2 and H3K36me3 [304]. These substrates include widely interspaced zinc finger motifs protein (WIZ), chromodomain Y-like protein (CDYL1), Cockayne syndrome group B protein (CSB, four peptides) and Histone-lysine N-methyltransferase (EHMT2). These substrates of KDM4 family are all members of transcription repression complexes, suggesting an additional role of transcriptional regulation by these demethylases. Using in vitro assays, KDM3A, KDM4E, KDM5C, and KDM6B have been shown to possess arginine demethylase activity, in addition to their lysine demethylase activity on peptide substrates [305]. Non-specific hydroxylase activities are not unusual for JmjC enzymes, so any such reported activity should demonstrate specificity and biological relevance, though it is possible that it will emerge that many lysine demethylases also possess relevant arginine demethylase activity.

Finally, a potentially new and exciting mechanism involves an epigenetic mark in RNA, methylation of 6 adenosine (m6A). mRNA m6A modification at the 5’UTR can recruit m6A binding proteins such as YTH N6-Methyladenosine RNA Binding Protein 2 (YTHDF2) in heat shock stress, which upregulates translation through binding eIF3 and the 40S subunit [306], and such a mechanism may prove true in other stress responses, such as hypoxia. The RNA demethylating enzymes FTO Alpha-Ketoglutarate Dependent Dioxygenase (FTO) and AlkB Homolog 5, RNA Demethylase (ALKBH5), are 2-OGDs, and although their ability to sense oxygen has yet to be investigated, their inhibition in hypoxia could result in an increase in global RNA methylation, which could regulate translation and RNA fate in hypoxia.

## 6. Relevance to Human Biology and Health

Although we currently do not know the importance of all of the 2-OGDs present in the genome, several of the key players in the hypoxia response have important functions and relevance to human biology and health. This is exemplified by the phenotypes observed in null mice, or by the presence of disease associated mutations in humans.

### 6.1. PHD/HIF/VHL Axis

Given the cellular functions mentioned above, it is no surprise that genetic mutations of most dioxygenases and HIFs have been implicated with human diseases. Mutations in *HIF-2α* and *PHD2* have been found in patients with vascular pathologies, such as erythrocytosis, polycythemia, and pheochromocytoma (Table 2). As HIF mediates hypoxia adaptation responses, including the regulation of erythropoiesis and vasculogenesis, it is not surprising that mutations within the *PHD*/*HIF*/*VHL* axis are associated with vascular pathologies. The crucial role of HIFs in vascular pathologies is strongly demonstrated by genetic studies of mice, as highlighted in Appendix A. Knockout mice of *HIF-1α* or *HIF-2α* are embryonic lethal with vascular defects (Table 2, Appendix A), whereas the deletion of *PHD2* that activates HIF signalling results in embryonic lethality in mice due to placental and heart defects (Table 2, Appendix A). *VHL* mutations also result in highly vascularized tumours, including pheochromocytomas, renal cell cancer carcinoma, retinal and central nervous system hemangioblastomas (Table 2). Hundreds of *VHL* mutations have been identified in VHL syndrome patients (listed in the Human Gene Mutation Database [307]). The homozygous *VHL* mutation R200W, which prevents efficient HIF-α degradation in normoxia, is found in all individuals with Chuvash Polycythemia (CP) [308]. CP is characterized by congenital erythrocytosis, and patients have been associated with pulmonary hypertension, thrombosis, vertebral hemangiomas, cerebral vascular events and other vascular abnormalities [309,310], displaying the role of VHL in HIF-dependent regulation of vasculogenesis and erythropoiesis.

### 6.2. 2-OGDs—Hydroxylases

Similar to PHDs, prolyl-4-hydroxylases P4HA1 and P4HA2 are hypoxia-inducible, but P4HA1/2 prolyl hydroxylation is required for different processes, that is collagen fiber formation. Consistent with their roles in collagen synthesis, *P4HA1* and *P4HA2* mutations are found in patients with collagen-related extracellular matrix disorders (Table 2; Appendix A). Furthermore, homozygous deletion of *P4HA1* is embryonic lethal with base membrane rupture due to defective collagen IV assembly (Table 2). PAHX is another hydroxylase, but of phytanoyl-CoA; essential for breaking down phytanic fatty acid. Mutations in *PAHX* is well associated with Refsum disease, a rare inherited neurological disorder caused by neurotoxic phytanic acid as these mutations result in an enzymatically inactive protein, thus leading to phytanic acid accumulation.

The roles of *TET1–3* in development are demonstrated in knockout mouse models (reviewed in [311]). *TET1*-null mice present several defects but these depend on the mode of genetic deletion. *TET3* deletion results in neonatal lethality, highlighting TET3 role in development. *TET3* mutations have been found in patients with intellectual disability and/or delayed global development (Table 2; Appendix A). Although somatic alterations of *TET2* have been found in several cancers, these mutations are majorly associated with myelodysplastic syndromes (Table 2; Appendix A). In addition to the listed mutations in Appendix A, a study reported *TET2* somatic mutations in 46 patients with myelodysplastic syndromes, myeloproliferative disorder, secondary acute myeloid leukemia, or chronic myelomonocytic leukemia [312]. Most of these mutations are predicted to lead to partial or total loss of function due to protein truncation.

### 6.3. 2-OGDs—JmjC Demethylases

Many of the JmjC demethylase genes have been associated with human diseases. In particular, several of them are mutated in patients with neurodevelopmental disorders, midline defects and cancers (Table 2; Appendix A). Although *KDM3A* is found to be mutated in infertile males [125], its role in infertility is not clear. Mutations in *KDM3B* are frequently implicated with intellectual disability, but also found in cancers including myeloid leukemias (Table 2). Similarly, *JMJD1C* mutations have been identified in individuals with autism spectrum disorder and intellectual disability. *JMJD1C* is also associated with congenital heart disease manifestation in 22q11.2 deletion syndrome patients. Amongst KDM4s, there are only two reports—single nucleotide substitutions of *KDM4C* in upper aerodigestive tract cancer and age of menarche. Mutations of *KDM5B*, *KDM5C* and *KDM6B* have been associated with neurodevelopment and a global developmental delay (Table 2; Appendix A). In particular, *KDM5C* is well recognised as an X-linked intellectual disability gene that is highly expressed in neural tissue. Mutations in PHD Finger Protein 8 (*PHF8*) are also associated with X-linked mental retardation and often accompanied by cleft lip/palate or autism. The phenotypes of *KDM5C* or *PHF8* mutations in humans are reflected by the deletion of these genes in mice (Table 2). On the other hand, *KDM6A* mutations are frequently found in individuals with Kabuki syndrome (KS), a genetic disease with developmental delay and congenital anomalies, (Table 2) highlighting the role of *KDM6A* in development. In addition to mutations listed in Appendix A, others have reported gross deletions, gross duplications, or chromosomal rearrangement in patients with KS or KS-like clinical manifestations [176,178,313,314,315]. However, whether the phenotypes observed are due to loss of demethylase activity solely is currently unknown.

Oxygen regulation is pivotal for embryogenesis as most developmental processes during early embryo development occurs in a hypoxic environment [316]. Given the fact that 2-OGDs sense and respond to oxygen levels, it is not surprising that mutations affecting the activity of several of these enzymes are detrimental to embryo development, contributing to neonatal defects. The hypoxic environment is also a common feature of tumours due to aberrant vascularisation and an inadequate supply of blood [317]. Mutations or deletions of some of these 2-OGDs are implicated in cancers. The molecular mechanisms through which 2-OGDs promote cancers is not completely understood. However, altered metabolism in cancer has been linked to alteration of 2-OGDs function (reviewed in [318]). 2-OGDs can be inhibited by metabolites that are accumulated in tumours, including succinate, fumarate and D/L-2-hydroxyglutarate [318,319]. It is thus very likely that the phenotypes observed in deletions and mutations are the result of both alteration of the hypoxia response and changes in metabolism. Overall, the presence and connections of HIFs or dioxygenase mutations in human disorders and the knockout studies demonstrate the essential roles of these genes.

## 7. Conclusions and Future Perspectives

As our understanding of the cellular response to hypoxia advances, new aspects continue to unravel. The role of oxygen has surfaced as far broader than just an acceptor molecule in oxidative phosphorylation in the mitochondria. Through acting as a co-factor for diverse and functionally important enzymes, oxygen is mechanistically identified as a potent signalling molecule in cells. The emerging focus of the field includes new aspects of chromatin regulation, RNA biology and broad regulation of protein post-translational modifications directly controlled by oxygen levels. This advanced understanding in conjunction with development of novel therapeutic chemicals targeting dioxygenases should provide not only exciting new biological insights, but also better treatments for patients suffering from a range of diseases. One area of technological advancement that will greatly progress the field is the adaptation of novel and unbiased quantitative techniques for measuring chromatin structure, transcriptional output, proteomic changes and cellular behavior. These approaches may provide resolution to some of the persisting major questions pertaining mechanisms controlling gene expression in response to hypoxia.

## Figures and Tables

**Figure 1 cancers-13-00350-f001:**
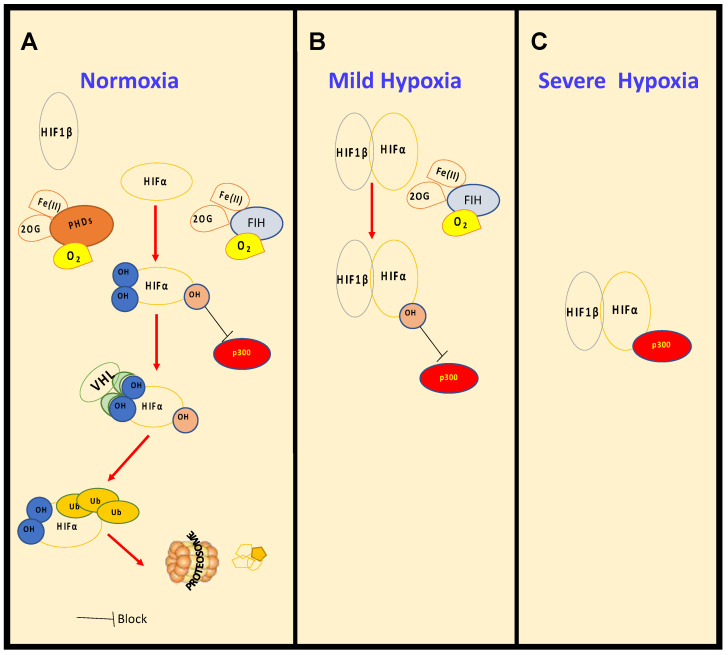
**Regulation of HIF levels and activity in normoxia and hypoxia.** Under normal oxygen conditions, (**A**), normoxia, HIF-α is constantly hydroxylated by PHDs and FIH. PHD-mediated hydroxylation increases binding affinity with the tumour suppressor VHL, which promotes ubiquitination and degradation by the proteosome. As oxygen levels decrease, in mild hypoxia (15–1% O_2_) (**B**), PHDs are inhibited, HIF-α is stabilised, though still hydroxylated by FIH, binds to HIF-1β and is able to induce transcription of certain target genes. With further reduction in oxygen levels, in severe hypoxia (<1% O_2_) (**C**), FIH is also inhibited and HIF is able to become fully active by the recruitment of co-activators such as p300.

**Figure 2 cancers-13-00350-f002:**
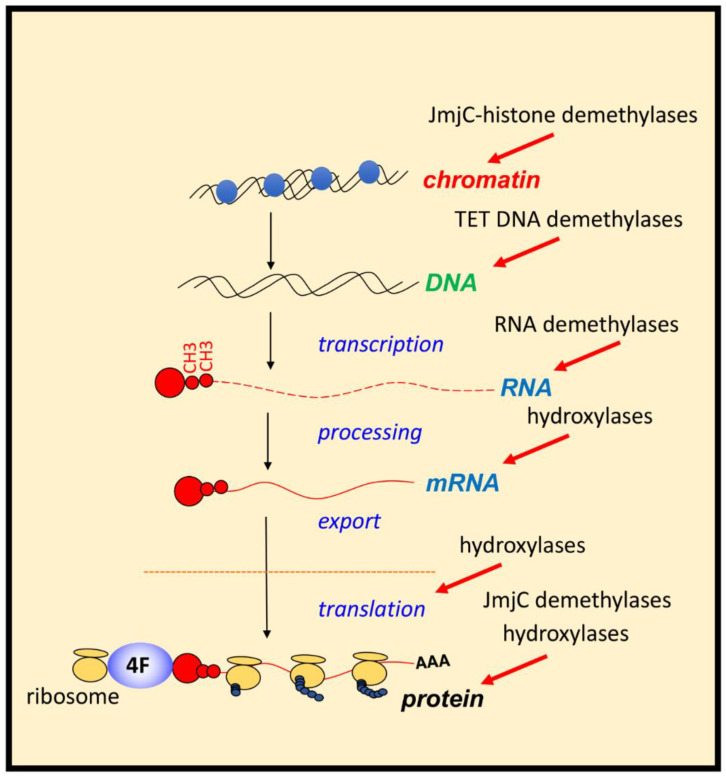
**Control of gene expression by 2-ODGs in hypoxia.** Hypoxia via the regulation of 2-OGDs has the potential of controlling all aspects of gene expression and protein function. Action of JmjC-histone demethylases and TETs (mediators of DNA demethylation) will impact on chromatin and DNA regulation. RNA demethylases, and several hydroxylases acts on mRNA processing, and fate. Hydroxylases also control rates of translation and ribosome activity, while JmjC-demethylases and hydroxylases can control protein function directly or indirectly by controlling other PTMs.

**Figure 3 cancers-13-00350-f003:**
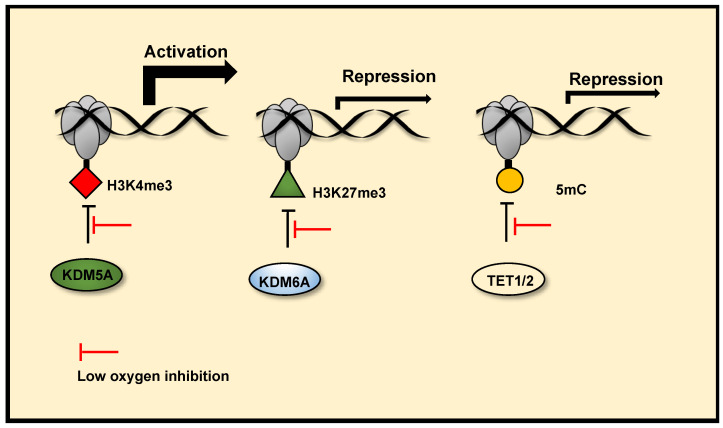
**Chromatin oxygen sensing via JmjC histone demethylases and TETs.** JmjC histone demethylases and TETs (mediators DNA demethylation) are 2-OGDs. Reduced activity of these enzymes in hypoxia, due to their oxygen sensitivity, can alter the chromatin landscape and mediate hypoxia induced transcription changes. Reduced activity of KDM5A in hypoxia increases H3K4me3 at the promoters of a subset hypoxia induced genes, facilitating their transcriptional activation. Reduced activity of KDM6A in hypoxia increases H3K27me3 at the promoters of a subset of hypoxia-repressed genes and represses their transcription. Reduced TET1/2 activity in hypoxia also represses gene transcription via DNA hypermethylation at gene promoters.

**Figure 4 cancers-13-00350-f004:**
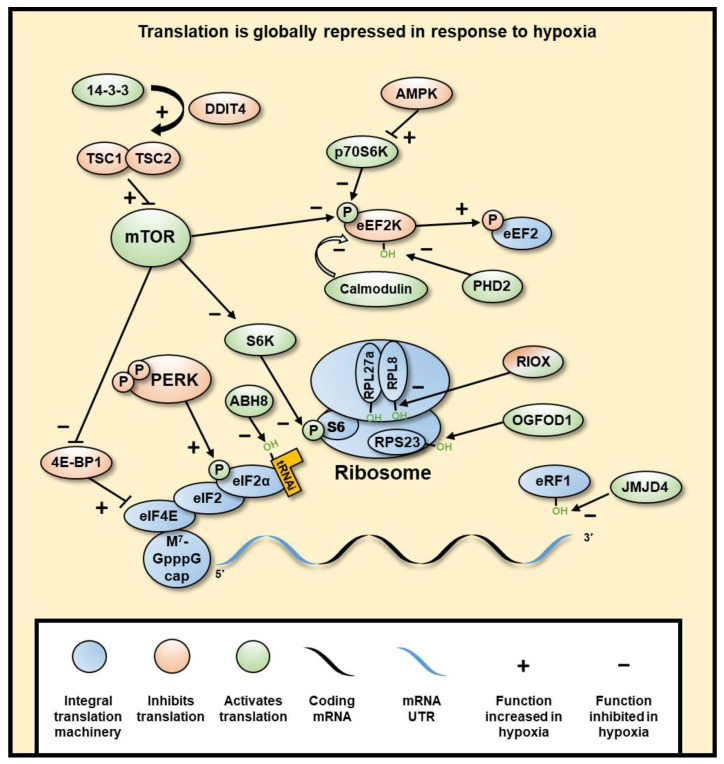
**Hypoxia induces a global inhibition of protein translation.** Global translation is mostly inhibited at initiation through mTOR inhibition of eIF4 and PERK inhibition of eIF2α, subsequently decreasing cap-dependent translation of mRNA. mTOR is inhibited through hypoxia-induced DDIT4-dependent release of TSC2 from 14-3-3 binding proteins, resulting in the inhibition of mTOR by TSC1/2 dimer. Elongation is also regulated through mTOR, as well as AMPK through its inhibition of RPS6KB1, wherein hypoxia eEF2K is not inhibited, which allows its phosphorylation and inhibition of eEF2. PHD2 can also hydroxylate eEF2K, which in normoxia causes its disassociation from calmodulin, decreasing its autophosphorylation. Termination is regulated by JMJD4-mediated hydroxylation of eRF1, which is required for termination. Selective translation of genes in hypoxia is regulated by UTR sequences, such as IRES and uORFs, which allow increased translation specifically in hypoxia. RNA binding proteins can bind to various parts of the mRNA and result in different regulatory outcomes. The ribosome can also be hydroxylated by RIOX1 and RIOX2, though it is not yet clear what role these modifications have.

**Figure 5 cancers-13-00350-f005:**
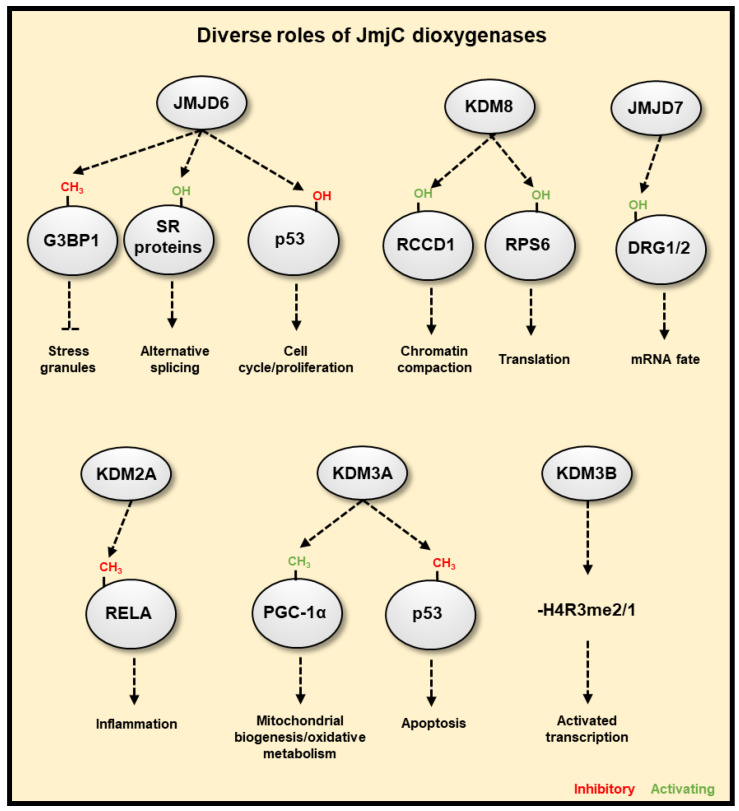
**JmjC-domain containing 2OG dioxygenases have diverse cellular functions.** JMJD6 is able to promote the formation of stress granules, which reversibly pauses transcript translation through demethylation of G3BP1. JMJD6 can hydroxylate splicing regulatory (SR) proteins, resulting in differential splicing or exon choice, such as skipping the first exon with the hydroxylation of SRSF11. JMJD6 can also hydroxylate p53 repressing its activity. KDM8 is an arginine hydroxylase that can target RCCD1 for chromatin condensation, and RPS6, which affects translation. JMJD7 is a lysyl hydroxylase that can target DRG1/2, affecting their regulation of RNA. KDM2A represses NF-κB activity via demethylation of RELA. KDM3A can demethylate both PGC-1α, promoting mitochondrial biogenesis, and p53, inhibiting its proapoptotic activity. KDM3B is able to demethylate mono- or di-methylated arginine residues, which functions to activate gene transcription.

**Table 1 cancers-13-00350-t001:** **2-OGDs with reported affinities for oxygen from in vitro assays.** The K_M_ values for oxygen have been determined via a range of in vitro experiments. The K_M_ obtained is affected by substrate, conditions, and technique, and therefore provides only a possible indication of oxygen sensing capability. We have assessed, from the reported K_M_s and cellular data, whether these 2-OGDs might act as oxygen sensors. * Median K_M_ from multiple studies (individual values listed in Appendix A).

2-OGD Type	Enzyme	O_2_ K_M_ (µM)	Potential O_2_ Sensor (Yes/No)	Substrate	Effect on Gene Expression in Hypoxia
Hydroxylases	PHD1	230 [17]	Y	Multiple	Y
	PHD2	240 * [12,17,18,19]	Y	Multiple	
	PHD3	230 [17]	Y	Multiple	
	4-PHα1	40 [17]		Collagen	
	PAHX	93 ± 43 [12,17]	Y	Isovaleryl CoA	
	CDO1	76 ± 17 [12]	N	Taurine	
	FIH	110 ± 30 * [12,20,21]	Y/N	Multiple	Y
Hydroxylases (known as DNA demethylases)	TET1	30 ± 10 [22]	N	DNA	Y
	TET2	30 ± 3 [22]	N	DNA	Y
JmjC demethylases	KDM4A	60 ± 20 * [23,24,25]	Y	H3K9me2/me3, H3K36me2, H1.4K26me2/me3	Y
	KDM4B	150 ± 40 [25]	Y	H3K9me2/me3, H3K36me2, H1.4K26me2/me3	Y
	KDM4C	158 ± 13 [24]	Y	H3K9me2/me3, H3K36me2, H1.4K26me2/me3Histone H3	Y
	KDM4E	197 ± 16 [24]	Y	H3K9me3Histone H3	Y
	KDM5A	90 ± 30 [25]	Y	H3K4me2/me3Histone H3	Y
	KDM5B	40 ± 10 [25]	N	H3K4me2/me3Histone H3	Y
	KDM5C	35 ± 10 [25]	N	H3K4me2/me3Histone H3	Y
	KDM5D	25 ± 5 [25]	N	H3K4me2/me3Histone H3	Y
	KDM6A	180 ± 40 [25]	Y	H3K27me2/me3Histone H3	Y
	KDM6B	20 ± 2 [25]	N	H3K27me2/me3Histone H3	Y
Cysteamine (2-aminoethanethiol) dioxygenase	ADO	>500 [26]	Y	RGS4 and RGS5	

**Table 2 cancers-13-00350-t002:** Available mice and human mutations phenotypes for HIF and dioxygenases.

Gene (Mouse/Human)	Homozygote Phenotype in Mouse	Human Phenotype
**HIFs**		
*Hif1a*/*HIF1A* (HIF-1α)	Embryonic lethal with cardiovascular malformations, cephalic vascularisation and neural tube defects [27,28,29]	Schizophrenia [30]. Maximal oxygen consumption [31]. Renal cell carcinoma [32].
*Epas1*/*EPAS1* (HIF-2α)	Embryonic lethal with bradycardia due to defective catecholamine homeostasis [33], vascular remodelling defects [34], cardiac failure and neonatal lethal with respiratory failure [35].	Congenital heart disorder [36]. Autism spectrum disorder [37]. Pheochromocytoma/paraganglioma-polycythaemia [38,39,40,41,42,43,44] /somatostatinoma [45]. Erythrocytosis and polycythaemia with paraganglioma [40,41,42]. Erythrocytosis [46,47,48,49,50,51]. Pulmonary arterial hypertension [52].
*Hif3a*/*HIF3A* (HIF-3α)	Mice deficient of an alternative spliced protein of *HIF-3α*, NEPAS, are viable and develop enlarged right ventricular owing to impaired pulmonary remodelling [53].	NR
**2-OGDs—hydroxylases**		
*Egln2*/*EGLN2* (PHD1)	Viable [54].	Increased risk of hepatocellular carcinoma [55], lung cancer [56,57], gastric cancer [58], colorectal cancer [59]. Pheochromocytoma/paraganglioma-polycythemia [60].
*Egln1*/*EGLN1* (PHD2)	Embryonic lethal with severe cardiac and placental defects [54].	High-altitude adaptation [45]. Erythrocytosis [61,62,63,64,65,66,67,68,69,70,71]. Pheochromocytoma/paraganglioma-polycythemia [60]. Pheochromocytoma [39]. Cardiopulmonary [72].
*Egln3*/*EGLN3* (PHD3)	Viable [54] with developmental defect of sympathoadrenal system [73].	NR
*P4ha1*/*P4HA1*	Embryonic lethal with delayed development and defective collagen IV assembly, resulting in base membrane rupture [74].	Congenital-onset disorder of connective tissue [75].
*P4ha2*/*P4HA2*	Viable and fertile with no obvious phenotypic abnormalities [76].	High myopia [77].
*Phyh*/*PHYH* (PAHX)	Viable without distinct developmental abnormalities [78].	Refsum disease [79,80,81,82,83,84,85]. Nonsyndromic cleft lip and palate [86].
*Hif1an*/*HIF1AN* (FIH)	Abnormal energy metabolism with reduced body weight, elevated metabolic rate and hyperventilation [87].	Colorectal cancer [88].
**2-OGDs—hydroxylases (mediators of DNA demethylation)**		
*Tet1*/*TET1*	Knockout of *TET1* via 5’ coding sequence results in partial embryonic lethality in mice [89,90,91], with surviving female mice displaying decreased fertility and reduced ovary size due to meiotic abnormality [89,90]. Whereas, mice with knockout via deletion of the catalytic domain of *TET1* are viable and fertile [91,92,93], with slightly reduced body size [92], as well as impaired spatial learning and short-term memory [94].	NR
*Tet2*/*TET2*	Disordered hematopoiesis and eventually develop myeloid malignancies [95,96,97], and T- and B-cell malignancies [96].	Myelodysplastic/myeloproliferative disease [98]. Prostate cancer [99]. Myeloproliferative neoplasms [100].
*Tet3*/*TET3*	Neonatal lethality [90,101].	Intellectual disability, developmental delay, autistic traits, hypotonia, growth abnormalities, facial dysmorphism and movement disorders [102].
**2-OGDs—hydroxylases (RNA demethylases)**		
*Fto*/*FTO*	Abnormal brain and cardiac development [103].	Developmental delay and dysmorphic facial features [104]. Growth retardation and multiple malformations [103]. Developmental delay and growth retardation [105]. Growth retardation and multiple malformations [106]. Obesity [107,108]. Type II diabetes [109]. Metabolic syndrome including obesity, hypertension, dyslipidemia, and defective glucose tolerance [110].
**2-ODGs—JmjC demethylases and hydroxylases**		
*Jmjd4*/*JMJD4*	Viable and fertile with normal physiology [111].	NR
*Jmjd6*/*JMJD6*	Perinatal lethal with growth retardation and exhibit severe tissue and organ differentiation defects, including brain, lung, liver, kidney, intestine, heart and thymus development at different stages of embryogenesis [112,113,114,115].	NR
*Kdm2a*/*KDM2A*	Embryonic lethal with severe growth retardation and defective neural tube closure [116].	NR
*Kdm2b*/*KDM2B*	*KDM2B-1*-deletion mice display moderate penetrance of neural tube defects, leading to exencephaly and death at birth [117]. Whereas, mice deficient of both *KDM2B-1* and *KDM2B-2* isoforms are embryonic lethal with fully penetrant developmental defects, including abnormal somitogenesis, reduced size, defective neural tube and heart [118,119,120]; especially a more severe developmental defect in female embryos [119]. Furthermore, KDM2B-2-deleted mice display similar developmental abnormalities with increased lethality, particularly in females [119].	NR
*Kdm3a*/*KDM3A*	Develop obesity, abnormal fat metabolism [121,122], reduced energy expenditure, and display metabolic syndrome, including, high plasma cholesterol, insulin, triglyceride, and leptin levels [122]. Male infertility, smaller testes and severe oligozoospermia [123]. Retarded mammary gland ductal growth in female knockout mice [124].	Male infertility [125].
*Kdm3b*/*KDM3B*	Postnatal growth restriction and female mice were infertile due to decreased ovulation, prolonged estrous cycles, reduced fertilisation and uterine decidual response [126]. Male knockout mice have impaired reproductive function, sperm development and maturation [126]. Knockout mice exhibit myelodysplastic syndrome and defective hematopoiesis including leukocytosis, moderate anemia, and granulocytosis [127].	Schizophrenia [128]. Intellectual disability [129]. Wilms tumour and hyperpigmentation [130]. Hepatoblastoma, autism, intellectual disability, and abnormal pigmentation [130]. Acute myeloid leukemia, mild intellectual disability, congenital hypothyroidism and congenital hip dysplasia [130]. Hodgkin lymphoma, feeding difficulties, intellectual disability, umbilical and inguinal hernia [131]. Intellectual disability, facial dysmorphism and short stature [131].
*Jmjd1c*/*JMJD1C*	Males gradually develop infertility with decreasing testes size due to progressive loss of germ cells [127].	Congenital heart disease in patients with 22q11.2 deletion syndrome [132]. Rett syndrome [133,134]. Autism spectrum disorder [133]. Intellectual disability [133]. Intracranial germ cell tumour [58].
*Kdm4a*/*KDM4A*	Viable [135].	NR
*Kdm4b*/*KDM4B*	Viable [136]. Viable with lower birth rate. Early weaning results in death. Susceptible to obesity with impaired energy expenditure, adaptive thermogenesis and adipose tissue lipolysis [137].	NR
*Kdm4c*/*KDM4C*	Viable and fertile [138]. However, another reported that it leads to embryonic lethality [139].	Upper aerodigestive tract cancer [140]. Age at menarche [141].
*Kdm4d*/*KDM4B*	Viable and fertile without gross abnormalities [142].	NR
*Kdm5a*/*KDM5A*	Viable [143,144]. Mice displayed mild behavioural and haematological abnormalities [143].	Intellectual disability [145]. Congenital heart disease [146].
*Kdm5b*/*KDM5B*	Embryonic lethal [147,148]. Neonatal lethal due to failure to establish respiratory function, defective neural system and homeotic skeletal transformations [149].	Intellectual disability, dyslexia, global developmental delay, facial dysmorphism, aggressive behaviour, hypospadias [150].
*Kdm5c*/*KDM5C*	Hemizygous *KDM5C* null male mice are embryonic lethal due to defective neurulation and cardiogenesis [151]. Male hemizygous knockout mice (*Kdm5c^−/y^*) Viable with adaptive and cognitive abnormalities, including increased aggression, impaired social behaviour, limited learning, fear memory deficits, defective dendritic spines [152,153] and significant reduced body weight [153].	X-linked intellectual disability [154,155,156,157,158,159,160,161,162,163,164,165,166]. Autism spectrum disorder [167].
*Kdm5d*/*KDM5D*	A large scale screening using CRISPR/Cas9-mediated genome editing reveals normal reproductive system in hemizygous *KDM5D*-knockout male mice [168].	NR
*Kdm6a*/*KDM6A*	Embryonic lethal with cardiac development defects and neural tube closure. While female knockout mice died mid-gestational, some hemizygous *KDM6A*-null male mice survive into adulthood [151,169,170,171,172] and are fertile [171,172], with reduced lifespan and smaller size [171]. Female embryonic lethal, abnormal/truncated posterior bodies, anaemic (hematopoiesis), severe heart development defect and neural tube closure. Male died around birth due to neural tube closure defect and inability to breath [173].	Kabuki syndrome [174,175,176,177,178,179,180,181,182,183]. Biliary atresia with Kabuki syndrome-like features [184]. Renal cancer [185].
*Kdm6b*/*KDM6B*	Embryonic [186] and perinatal lethal with respiratory failure [187,188,189], detail reviewed here [190]. Reduced proliferation and hypertrophy of chondrocytes, as well as delayed endochondral ossification in mice [191]. Delayed osteoblast differentiation and bone ossification [192].	Intellectual disability [145]. Intellectual disability, brachydactyly and dysmorphism [193].
*Uty*/*UTY*	Hemizygous male mice are viable [169].	NR
*Kdm7a*/*KDM7A*	A large-scale genome-wide tissue phenotype screen revealed that abnormal hair follicles, sebaceous gland, tail and hair follicle bulge morphology in KDM7A knockout mice [194].	
*Phf8*/*PHF8*	Impaired learning and memory, hippocampal long-term potentiation [195].	X-linked mental retardation with cleft lip/palate [196,197,198]. Autism and Asperger syndrome [199]. Autism spectrum disorder, intellectual disability, cleft palate and Aarskog syndrome [200]. Intellectual disability [159].
*Kdm8*/*KDM8*	Embryonic lethal with delayed development in multiple organs [201] and growth retardation [202].	NR

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
