# Peer review of "Roles of HIF and 2-Oxoglutarate-Dependent Dioxygenases in Controlling Gene Expression in Hypoxia"

_cancers, 2021, doi:10.3390/cancers13020350_

Round 1
Reviewer 1 Report
This is a very nice and comprehensive review on the hypoxia, oxygen sensing and transcription regulation. The information reviewed here will be very helpful for the community. I have a couple of minor points.
- For Line 402-405, this also needs to add the hydroxylation of ADSL by PHD1 (PMID: 31729379), TBK1 by PHD2 (PMID: 31810986) and SFMBT1 by PHD2 ( PMID: 32023483).
Fig.6 may need some clarification. JMJD6 was reported to hydroxylate p53 lysine residue. KDM8 was reported to repress p53 function, not sure whether had anything to do with p53 hydroxylation
Author Response
We thank the reviewer for the comments and have made the suggested changes to the text. We added the suggested references and clarified Figure 6, now figure 5.
Reviewer 2 Report
The article by Frost et al. entitled “Roles of HIF and 2-Oxoglutarate dependent enzymes in controlling gene expression in hypoxia” summarizes the roles of both HIFs and iron(II)/2-oxoglutarate-dependent dioxygenases (2-OGDs) in coordinating hypoxia response by regulating each level gene expression: from directly modulating chromatin accessibility to direct modulation of non-HIF and non-histone proteins that are involved in gene expression. The manuscript is nicely written and provides rich information for readers and researchers looking to learn more about the involvement of 2-OGDs in hypoxia-responsive gene expression, especially those 2-OGDs outside the realm of the highly studied prolyl hydroxylase family. Another concern that I have is that although the article is well-cited, several highly relevant studies were not included in the manuscript (I will list specific studies below), some of which would not support the traditional ‘oxygen sensing’ narrative that the authors are trying to convey. It is important that the readers are made aware of such anomalies in the field. Nonetheless, I am in favor of publishing this review article, but I have suggestions to first address this concern and several other suggestions to help focus the article and clarify specific aspects of the text and figures to the readers.
Major revisions
- The roles that HIF plays in controlling gene expression in hypoxia is a well-reviewed topic and therefore the text pertaining to this should be significantly trimmed or be completely removed. Specifically, I suggest removing or significantly trimming the text from lines 92-195 (section titled “[H1] Effects of hypoxia on gene transcription”). The novelty of this review article lies in the latter half of the article title. The primary introductory text from lines 34-89 provides an excellent introduction to this topic. Lastly, significantly trimming or removal of lines 92-204 should be accompanied by a change in the article title to reflect the focus on the roles of 2-OGDs in controlling gene expression in hypoxia.
- I have several concerns and suggestions regarding “Box1.” outlined below.
- Given the numerous studies reporting KM values for PHD2 and FIH, it is reasonable that the authors provide a median KM For complete transparency it is suggested that the authors provide a supplementary table listing the individual KM values that were used to derive the median value and provide citations within the table.
- Within Box1 the authors should provide citations to the studies from which these KM values were derived.
- The authors should specify the error associated with the KM values (e.g., for KDM5A the KM,O2 is 90 ± 30 μM).
- At least for the lysine demethylases (KDM4s, KDM5s, and KDM6s), the authors should be more specific in the “Substrate” column to provide complete clarity. For example, KDM4A has been shown to demethylate di- and tri-methylated Histone H3 at lysine-9 and lysine-36, thus simply stating “Histone H3” is ambiguous.
- For KDM4A, three individual KM values have been reported in the literature and all three should be included in Box1 (57 ± 10 μM, 60 ± 20 μM, 173 ± 23 μM). The PMIDs for these studies are PMID: 30872525, PMID: 23067339, and PMID: 28051298. The readers should be aware of this discrepancy in oxygen KM values for KDM4A to not convey that KDM4A is an established oxygen sensor. This should also be touched on within text where appropriate.
- Maybe I am missing something here, but the criteria for whether an enzyme is denoted as an oxygen sensor or not within the “Potential O2 sensor (Yes/No)” column is not clear. Specifically, FIH KM,O2 110 μM is denoted as “Y/N” whereas the lower KDM5A KM,O2 90 ± 30 μM and PAHX KM,O2 93 μM are denoted as only “Y”. Perhaps this column is referring to whether there is evidence of oxygen sensing in cell models? Nonetheless, the authors should clarify the reasoning behind the “Y/N” assignments.
- The information provided within the figures is sometimes ambiguous, confusing, or lacking key details. My specific suggestions are provided below.
- Figure 1: Within Panel A the authors should show two proline hydroxylation events, rather than one, to not suggest there is only one proline hydroxylation event. Additionally, the authors should illustrate that the hydroxylated asparagine residue (Panels A and B) blocks association with co-activators.
- Figure 2: It is not clear what the arrow stemming from “hydroxylases” is pointing towards. It appears it is pointing towards the process “translation” but should be pointing towards macromolecules involved in translation (“protein” or “tRNAPhe”).
- Figure 3: It is suggested that this figure be removed from the article as the contents are more than well-reviewed in literature. In place of this figure, it is suggested that the authors build either figure or table that informs the reader of which 2-OGDs are upregulated in hypoxia, and/or direct transcriptional targets of HIFs.
- Figure 4: The simultaneous use of “T-shaped” arrow on the right-hand side of the figure and the use or regular arrows on the left-hand side of the figure introduces some unneeded complexity. The 2-OGDs shown in this figure, are inhibitory of the modifications shown. Thus only “T-shaped” arrow should be used within the full figure. The authors should directly illustrate that hypoxia inhibits the function of these 2-OGDs (perhaps with an adjacent “T-shaped” arrow perpendicular to the ones stemming from the enzyme). In that regard, the “O2 levels” gradient at the top of the figure is not needed.
- Figure 5: the authors are illustrating that PHD2 inhibits the eEF2K hydroxylation event with the use of a “T-shaped” arrow, whereas it should be shown that PHD2 imparts the modification upon eEF2K with the use of a regular arrow. Furthermore, the use of solid vs dashed lines to denote “function in hypoxia” or “function prevented in hypoxia” introduces some confusion. Immediately I am confused whether the solid lines are denoting processes that are (1) enhanced in hypoxia or (2) those that continue to occur in hypoxia to the same extent as in normoxia. It is suggested that all lines are made solid, and any inhibitory or enhancing effects of hypoxia are shown within the figure (perhaps with plus or minus symbols above the arrow tails). Lastly, the text described on lines 372-380 appears to be relevant to Figure 5, or at least the process of translation, and should be illustrated.
- Figure 6: It appears the authors are illustrating that KDM8 inhibits the hydroxylation of p53. The authors should adjust this figure if that statement is not what they had intended to convey. Additionally, it is not clear why the authors have illustrated the JMJD6 interaction with BRD4 and P-TEFb within this figure given that no hydroxylation event has been described for these two interactions within the text. It is suggested that the authors only include within Figure 6 those interactions that occur within the context of 2-ODG enzymatic activity. The inclusion of non-enzymatic modifying protein-protein interactions (PPIs) opens this figure to include the tens-to-hundred of PPIs that occur between lysine demethylases are their respective interacting proteins. Lastly, the text on lines 456-460 appears relevant to Figure 6 and should be illustrated.
- The statement “Future work should investigate if the oxygen-sensitive H3K9me3 demethylase activity of KDM4A is linked to controlling of gene expression and chromatin regulation in hypoxia” (lines 259-261) hasn’t considered work from others such as Lee et al. (PMID: 23884959).
- From lines 261-263 it appears as though the authors are alluding to the idea that it is well-established that KDM3A retains activity in hypoxia. Though this is supported by the cited literature, it is also crucial to consider the work of others than have demonstrated KDM3A inhibition in hypoxia (PMID: 20881000, PMID: 31629659).
- The discussion of non-HIF PHD substrates from lines 402-408 is relevant, though should be followed by the mention of recent work by Cockman et al. (PMID 31500697) to make the readers aware of the current anomalies in the field.
- The paragraph beginning on line 448 describing “KDM8”, or more appropriately named JMJD5, is somewhat misleading as it suggests this JmjC hydroxylase is an established arginine hydroxylase as there is no mention of the other activities displayed by this enzyme. There is evidence for peptidase activity, arginine hydroxylation, and lysine demethylation, of which the last mentioned is controversial. I suggest the authors make the readers aware of this. The UniProt page for this enzyme (ID: Q8N371) provides a decent summary of the activities displayed by this enzyme. Similarly, the peptidase activity displayed by JMJD7 should also be mentioned.
- The paragraph beginning on line 456 appears to focus on discussing non-histone or non-canonical targets of lysine demethylases. This is highly relevant, but the authors have not comprehensively covered this area of research. It is suggested the authors include in-text describing the non-canonical substrate of KDM3B (PMID: 29641999), reported in vitro non-histone substrates of KDM4s (PMID: 19799855), and reported non-histone substrates of KDM3A (PMID: 31629659; PMID: 27270439). Finally, it is suggested that the authors end this paragraph by referring to the implications of the works by Walport et al. (PMID: 27337104).
- The study by Qian et al. (PMID: 31629659) is highly relevant to the author's topic and should be discussed in the main section where appropriate.
Minor revisions
- The article title should be modified to specifically convey the focus on 2-oxoglutarate-dependent dioxygenases specifically, to not confuse with other types of 2-oxoglutarate-dependent enzymes.
- On line 29 “2-OG dioxygenases” should read as “2-OGDs”.
- On line 30 “…and their relevance to human cancers” should read as “and their relevance to human biology and health” to better reflect the contents of the last section starting on lines 461.
- On line 42 it is suggested to remove “(encoded by the EPAS1 gene)” given that the gene names for the other two isoforms are not provided. Additionally, it suggested to directly specify that the three HIF-α isoforms are encoded by different genes.
- On line 54 “This modification is a proline hydroxylation” should read as “This modification is two separate proline hydroxylation events” to not imply one proline hydroxylation event occurs.
- On line 56 the authors should indicate that these enzymes require “iron (II)” to be more specific.
- On line 56 “activity.[8]. Mammals” should read as “activity [8]. Mammals”.
- On line 68 and 176 it is not clear what “[G]” is referring to.
- On line 73 Supplementary Table 1 is referred to, however, Table 1 appears to be more appropriate for a sentence that appears to focus on highlighting the roles of HIFs and 2-OGDs in development and disease.
- The authors should provide citations for the studies referred to between lines 73-76.
- On line 146 the text “Sharma, 2013 #656;Hoang, 2001 #293}” appears to be a minor error with a reference manager software.
- On line 232 “while other are firmly associated with closed conformation” should read as “while others are firmly associated with the closed conformation”.
- The discussion surrounding KDM4A oxygen sensitivity beginning on line 253 should also address the studies that are contradictory to this claim (i.e., those studies that have determined low KM,O2 values for this enzyme.
- The statement “This may provide a mechanism of increasing HIF-1α levels in conditions of hypoxia…” on lines 258-259 does not align with the logic of the previous sentence. The H3K9me3 modification at the HIF-1A gene is inhibitory of gene expression, thus a loss of KDM4A H3K9me3 demethylase activity in hypoxia would be anticipated to lead to a reduction in HIF-1α levels, rather than an increase as the authors have stated. It is suggested the authors change the wording to “This may provide a mechanism of maintaining HIF-1α levels in conditions of mild hypoxia, specifically, …” to better reflect the study they have cited (Dobrynin et al., 2017, PMID: 28894274).
- Throughout the article, the writing somewhat suggests that 2-OGD inhibition in hypoxia only occurs due to a direct reduction in oxygen available to such enzymes (this is the only aspect discussed). However, it is well established that the landscape of metabolites and reactive oxygen species may shift in response to hypoxia confer inhibition of 2-OGDs. It is suggested that the authors make the readers aware of these additional avenues of 2-OGD inhibition in hypoxia where appropriate, perhaps in the introduction. An example of the metabolic aspect has been reviewed by Chang et al. (PMID: 31221981).
- On lines 319 “… the energy demands of the cell…” should read as “… the energy expenditure of the cell…”.
- The statement “Hydroxylation of splicing regulatory (SR) proteins results in differential splicing or exon choice, such as skipping the first exon with the hydroxylation of SRSF11” within the Figure 5 caption (lines 335-336) does not appear to be reflected within the figure but is more appropriate to include in the Figure 6 caption.
- On line 370 it is suggested that the authors re-word or remove the statement “JmjC hydroxylases hydroxylate histidyl residues in ribosomal proteins” to not convey that it is established that all JmjC hydroxylases have inherent histidyl hydroxylation activity, but only specific ones do.
- The text from lines 409-419 is irrelevant to the focus of the article on 2-OGDs and should be removed.
- On line 427 the authors should be more specific in regards to the type of arginine demethylation facilitated by JMJD6 (e.g., mono-methyl, di-methyl asymmetric, di-methyl symmetric?).
- On lines 431-437 the authors should state the specific residue positions on these target proteins that are modified by JMJD6 activity.
- The discussion surrounding JMJD6 on lines 424-447 should also discuss auto/self-hydroxylase activity which was reported to be important for JMJD6 function (PMID PMID: 22189873).
Author Response
The article by Frost et al. entitled “Roles of HIF and 2-Oxoglutarate dependent enzymes in controlling gene expression in hypoxia” summarizes the roles of both HIFs and iron(II)/2-oxoglutarate-dependent dioxygenases (2-OGDs) in coordinating hypoxia response by regulating each level gene expression: from directly modulating chromatin accessibility to direct modulation of non-HIF and non-histone proteins that are involved in gene expression. The manuscript is nicely written and provides rich information for readers and researchers looking to learn more about the involvement of 2-OGDs in hypoxia-responsive gene expression, especially those 2-OGDs outside the realm of the highly studied prolyl hydroxylase family. Another concern that I have is that although the article is well-cited, several highly relevant studies were not included in the manuscript (I will list specific studies below), some of which would not support the traditional ‘oxygen sensing’ narrative that the authors are trying to convey. It is important that the readers are made aware of such anomalies in the field. Nonetheless, I am in favor of publishing this review article, but I have suggestions to first address this concern and several other suggestions to help focus the article and clarify specific aspects of the text and figures to the readers.
We thank this reviewer for their constructive criticism. We have addressed the specific points below
Major revisions
- The roles that HIF plays in controlling gene expression in hypoxia is a well-reviewed topic and therefore the text pertaining to this should be significantly trimmed or be completely removed. Specifically, I suggest removing or significantly trimming the text from lines 92-195 (section titled “[H1] Effects of hypoxia on gene transcription”). The novelty of this review article lies in the latter half of the article title. The primary introductory text from lines 34-89 provides an excellent introduction to this topic. Lastly, significantly trimming or removal of lines 92-204 should be accompanied by a change in the article title to reflect the focus on the roles of 2-OGDs in controlling gene expression in hypoxia.
The content discussed in lines 92-204 have been removed, excepting discussion of m6A RNA demethylases, which as been relocated to the section now entitled “Other potential roles of 2-OGDs in the hypoxia response”
- I have several concerns and suggestions regarding “Box1.” outlined below.
Given the numerous studies reporting KM values for PHD2 and FIH, it is reasonable that the authors provide a median KM For complete transparency it is suggested that the authors provide a supplementary table listing the individual KM values that were used to derive the median value and provide citations within the table.
These values are now listed in Supplementary Table 1.
Within Box1 the authors should provide citations to the studies from which these KM values were derived.
These citations have been added as suggested.
The authors should specify the error associated with the KM values (e.g., for KDM5A the KM,O2 is 90 ± 30 μM).
Thank you for pointing this out to us. The error values have been added as suggested.
At least for the lysine demethylases (KDM4s, KDM5s, and KDM6s), the authors should be more specific in the “Substrate” column to provide complete clarity. For example, KDM4A has been shown to demethylate di- and tri-methylated Histone H3 at lysine-9 and lysine-36, thus simply stating “Histone H3” is ambiguous.
The detailed substrate information has been added for the lysine demethylases.
For KDM4A, three individual KM values have been reported in the literature and all three should be included in Box1 (57 ± 10 μM, 60 ± 20 μM, 173 ± 23 μM). The PMIDs for these studies are PMID: 30872525, PMID: 23067339, and PMID: 28051298. The readers should be aware of this discrepancy in oxygen KM values for KDM4A to not convey that KDM4A is an established oxygen sensor. This should also be touched on within text where appropriate.
This range of oxygen affinities are now listed in Supplementary Table 1, and referenced in the text with discussion of KDM4A’s oxygen sensitivity.
Maybe I am missing something here, but the criteria for whether an enzyme is denoted as an oxygen sensor or not within the “Potential O2 sensor (Yes/No)” column is not clear. Specifically, FIH KM,O2 110 μM is denoted as “Y/N” whereas the lower KDM5A KM,O2 90 ± 30 μM and PAHX KM,O2 93 μM are denoted as only “Y”. Perhaps this column is referring to whether there is evidence of oxygen sensing in cell models? Nonetheless, the authors should clarify the reasoning behind the “Y/N” assignments.
These assignments were indeed taking into account both the reported oxygen KM and biological evidence of their ability to sense oxygen. We admit that this was an oversimplification, and we have removed this from the manuscript.
- The information provided within the figures is sometimes ambiguous, confusing, or lacking key details. My specific suggestions are provided below.
Figure 1: Within Panel A the authors should show two proline hydroxylation events, rather than one, to not suggest there is only one proline hydroxylation event. Additionally, the authors should illustrate that the hydroxylated asparagine residue (Panels A and B) blocks association with co-activators.
We have added this additional information on the figure as requested
Figure 2: It is not clear what the arrow stemming from “hydroxylases” is pointing towards. It appears it is pointing towards the process “translation” but should be pointing towards macromolecules involved in translation (“protein” or “tRNAPhe”).
The arrows are pointing to general processes or molecules. Illustrating the breath of action of these enzymes.
Figure 3: It is suggested that this figure be removed from the article as the contents are more than well-reviewed in literature. In place of this figure, it is suggested that the authors build either figure or table that informs the reader of which 2-OGDs are upregulated in hypoxia, and/or direct transcriptional targets of HIFs.
We have removed this figure as suggested.
Figure 4: The simultaneous use of “T-shaped” arrow on the right-hand side of the figure and the use or regular arrows on the left-hand side of the figure introduces some unneeded complexity. The 2-OGDs shown in this figure, are inhibitory of the modifications shown. Thus only “T-shaped” arrow should be used within the full figure. The authors should directly illustrate that hypoxia inhibits the function of these 2-OGDs (perhaps with an adjacent “T-shaped” arrow perpendicular to the ones stemming from the enzyme). In that regard, the “O2 levels” gradient at the top of the figure is not needed.
We agree with recommendations for changes to figure 4 and have changed it accordingly. This is now figure 3
Figure 5: the authors are illustrating that PHD2 inhibits the eEF2K hydroxylation event with the use of a “T-shaped” arrow, whereas it should be shown that PHD2 imparts the modification upon eEF2K with the use of a regular arrow. Furthermore, the use of solid vs dashed lines to denote “function in hypoxia” or “function prevented in hypoxia” introduces some confusion. Immediately I am confused whether the solid lines are denoting processes that are (1) enhanced in hypoxia or (2) those that continue to occur in hypoxia to the same extent as in normoxia. It is suggested that all lines are made solid, and any inhibitory or enhancing effects of hypoxia are shown within the figure (perhaps with plus or minus symbols above the arrow tails). Lastly, the text described on lines 372-380 appears to be relevant to Figure 5, or at least the process of translation, and should be illustrated.
These changes have been made to figure 5, which is now figure 4.
Figure 6: It appears the authors are illustrating that KDM8 inhibits the hydroxylation of p53. The authors should adjust this figure if that statement is not what they had intended to convey. Additionally, it is not clear why the authors have illustrated the JMJD6 interaction with BRD4 and P-TEFb within this figure given that no hydroxylation event has been described for these two interactions within the text. It is suggested that the authors only include within Figure 6 those interactions that occur within the context of 2-ODG enzymatic activity. The inclusion of non-enzymatic modifying protein-protein interactions (PPIs) opens this figure to include the tens-to-hundred of PPIs that occur between lysine demethylases are their respective interacting proteins. Lastly, the text on lines 456-460 appears relevant to Figure 6 and should be illustrated.
These changes have been made to figure 6, now figure 5.
- The statement “Future work should investigate if the oxygen-sensitive H3K9me3 demethylase activity of KDM4A is linked to controlling of gene expression and chromatin regulation in hypoxia” (lines 259-261) hasn’t considered work from others such as Lee et al. (PMID: 23884959).
We have amended accordingly and added relevant info from PMID: 23884959 along with the reference.
- From lines 261-263 it appears as though the authors are alluding to the idea that it is well-established that KDM3A retains activity in hypoxia. Though this is supported by the cited literature, it is also crucial to consider the work of others than have demonstrated KDM3A inhibition in hypoxia (PMID: 20881000, PMID: 31629659).
We have cited the additional relevant literature and amended the text to point out that others have reported KDM3A inhibition in hypoxia.
- The discussion of non-HIF PHD substrates from lines 402-408 is relevant, though should be followed by the mention of recent work by Cockman et al. (PMID 31500697) to make the readers aware of the current anomalies in the field.
We have added the reference for discussion on non-HIF substrates.
- The paragraph beginning on line 448 describing “KDM8”, or more appropriately named JMJD5, is somewhat misleading as it suggests this JmjC hydroxylase is an established arginine hydroxylase as there is no mention of the other activities displayed by this enzyme. There is evidence for peptidase activity, arginine hydroxylation, and lysine demethylation, of which the last mentioned is controversial. I suggest the authors make the readers aware of this. The UniProt page for this enzyme (ID: Q8N371) provides a decent summary of the activities displayed by this enzyme. Similarly, the peptidase activity displayed by JMJD7 should also be mentioned.
This information has been incorporated into the text.
- The paragraph beginning on line 456 appears to focus on discussing non-histone or non-canonical targets of lysine demethylases. This is highly relevant, but the authors have not comprehensively covered this area of research. It is suggested the authors include in-text describing the non-canonical substrate of KDM3B (PMID: 29641999), reported in vitro non-histone substrates of KDM4s (PMID: 19799855), and reported non-histone substrates of KDM3A (PMID: 31629659; PMID: 27270439). Finally, it is suggested that the authors end this paragraph by referring to the implications of the works by Walport et al. (PMID: 27337104).
This paragraph has been expanded to include reference to and discussion of the suggested studies.
- The study by Qian et al. (PMID: 31629659) is highly relevant to the author's topic and should be discussed in the main section where appropriate.
This study has been discussed within the expanded discussion suggested in the previous point.
Minor revisions
- The article title should be modified to specifically convey the focus on 2-oxoglutarate-dependent dioxygenases specifically, to not confuse with other types of 2-oxoglutarate-dependent enzymes.
We have made the modification as suggested.
- On line 29 “2-OG dioxygenases” should read as “2-OGDs”.
We have corrected this.
- On line 30 “…and their relevance to human cancers” should read as “and their relevance to human biology and health” to better reflect the contents of the last section starting on lines 461.
We have made the suggested change.
- On line 42 it is suggested to remove “(encoded by the EPAS1 gene)” given that the gene names for the other two isoforms are not provided. Additionally, it suggested to directly specify that the three HIF-α isoforms are encoded by different genes.
We have made the suggested changes and also removed HIF-1βs gene name (ARNT) for consistency.
- On line 54 “This modification is a proline hydroxylation” should read as “This modification is two separate proline hydroxylation events” to not imply one proline hydroxylation event occurs.
We have made the suggested change.
- On line 56 the authors should indicate that these enzymes require “iron (II)” to be more specific.
We have made the suggested change.
- On line 56 “activity.[8]. Mammals” should read as “activity [8]. Mammals”.
We have made the suggested change.
- On line 68 and 176 it is not clear what “[G]” is referring to.
We have corrected this
- On line 73 Supplementary Table 1 is referred to, however, Table 1 appears to be more appropriate for a sentence that appears to focus on highlighting the roles of HIFs and 2-OGDs in development and disease.
We have corrected this now.
- The authors should provide citations for the studies referred to between lines 73-76.
These are now included in box 1
- On line 146 the text “Sharma, 2013 #656;Hoang, 2001 #293}” appears to be a minor error with a reference manager software.
We have made the correction.
- On line 232 “while other are firmly associated with closed conformation” should read as “while others are firmly associated with the closed conformation”.
We have made the suggested change.
- The discussion surrounding KDM4A oxygen sensitivity beginning on line 253 should also address the studies that are contradictory to this claim (i.e., those studies that have determined low KM,O2 values for this enzyme.
This range of oxygen affinities are now listed in Supplementary Table 1, and referenced in the text with discussion of KDM4A’s oxygen sensitivity.
- The statement “This may provide a mechanism of increasing HIF-1α levels in conditions of hypoxia…” on lines 258-259 does not align with the logic of the previous sentence. The H3K9me3 modification at the HIF-1A gene is inhibitory of gene expression, thus a loss of KDM4A H3K9me3 demethylase activity in hypoxia would be anticipated to lead to a reduction in HIF-1α levels, rather than an increase as the authors have stated. It is suggested the authors change the wording to “This may provide a mechanism of maintaining HIF-1α levels in conditions of mild hypoxia, specifically, …” to better reflect the study they have cited (Dobrynin et al., 2017, PMID: 28894274).
We have made the suggested change
- Throughout the article, the writing somewhat suggests that 2-OGD inhibition in hypoxia only occurs due to a direct reduction in oxygen available to such enzymes (this is the only aspect discussed). However, it is well established that the landscape of metabolites and reactive oxygen species may shift in response to hypoxia confer inhibition of 2-OGDs. It is suggested that the authors make the readers aware of these additional avenues of 2-OGD inhibition in hypoxia where appropriate, perhaps in the introduction. An example of the metabolic aspect has been reviewed by Chang et al. (PMID: 31221981).
We have added a sentence to reflect this fact, but our focus is on oxygen sensing for this review.
- On lines 319 “… the energy demands of the cell…” should read as “… the energy expenditure of the cell…”.
We have made the suggested change
- The statement “Hydroxylation of splicing regulatory (SR) proteins results in differential splicing or exon choice, such as skipping the first exon with the hydroxylation of SRSF11” within the Figure 5 caption (lines 335-336) does not appear to be reflected within the figure but is more appropriate to include in the Figure 6 caption.
This text has been moved to the legend of figure 5.
- On line 370 it is suggested that the authors re-word or remove the statement “JmjC hydroxylases hydroxylate histidyl residues in ribosomal proteins” to not convey that it is established that all JmjC hydroxylases have inherent histidyl hydroxylation activity, but only specific ones do.
We have re-worded this sentence to make sure that we are not inferring all JmjC hydroxylases have histidyl hydroxylation activity.
- The text from lines 409-419 is irrelevant to the focus of the article on 2-OGDs and should be removed.
We have deleted this part to focus more on 2-OGDs.
- On line 427 the authors should be more specific in regards to the type of arginine demethylation facilitated by JMJD6 (e.g., mono-methyl, di-methyl asymmetric, di-methyl symmetric?).
We have added the specific information.
- On lines 431-437 the authors should state the specific residue positions on these target proteins that are modified by JMJD6 activity.
We have added the specific modified residues.
- The discussion surrounding JMJD6 on lines 424-447 should also discuss auto/self-hydroxylase activity which was reported to be important for JMJD6 function (PMID PMID: 22189873).
We have added the discussion of the auto-hydroxylation of Jmjd6 as recommended.
Reviewer 3 Report
This is a comprehensive review on the role of oxygen-sensitive pathways in regulating a wide variety of biological programs, including physiologic hypoxic response, oncogenic programs, and translational control. Both HIF-dependent and HIF-independent pathways are highlighted. The review is well structured; however, there are some additional recommended clarifications. Some of these are recommended with the goal of helping non-expert readers; whereas, others illustrate some of the existing inconsistencies in the field.
Specific Comments:
1) Line 42: The section describing the 3 different isoforms would benefit by clarifying that HIF3a lacks a TAD, and thus either acts as a dominant-negative or transcription-independent regulator of hypoxic response.
2) Line 49 reads: “The mechanism leading to the activation of HIF was unravelled in 2001 [6,7]”. Although, these two seminal studies did indeed cement the mechanism of hypoxic HIF activation via prolyl hydroxylation, a series of previous findings had begun linking HIF proteolysis to pVHL in hypoxia [for example, Maxwell PH (1999) Nature and Ohh M (2000) Nat Cell Biol.]. Line 49 fails to capture this idea sufficiently.
3) Line 58: “Biochemical characterisation revealed that PHDs have low affinity for molecular oxygen.” This statement needs citations.
4) Line 73: “Furthermore, genomic techniques…in response to 2-OGD inhibition.” This statement needs citations to highlight the findings.
5) O2 Km values vary significantly between reports, perhaps because of differences in substrate lengths, buffer conditions, etc. As done with Table 1, it might help to put citations in Box 1 to allow readers to identify the experimental conditions under which a given Km was measured.
6) ADO can be added to Box 1 as an oxygen sensor. Masson N etal (2019) Science.
7) The role of KDM3A in hypoxia remains confusing. While the authors highlight (line 263) that KDM3A has been shown to support HIF function, and thus is likely to be active in hypoxia; Qian X etal (2019) Mol Cell reported that KDM3A is an oxygen-sensitive histone demethylase. This idea needs to clarified in the text.
8) Links between translational regulation and HIF extend beyond hypoxia. For example, links between HIF and XBP1 [Chen X (2014) Nature] could be highlighted.
9) There are a few instances of typographical errors. Some of these include:
- Line 68: extra [G] in, ‘chromatin [G] structure and epigenetics [G] to RNA biology’
- Line 176: extra [G] in, ‘The first discovered mechanism of selective translation was internal ribosomal entry sites (IRES) [G]’
- Line 415: SLC2A1 typo in, ‘Glucose transporter 1 (GLUT1) (gene name SLCA1)’.
Author Response
This is a comprehensive review on the role of oxygen-sensitive pathways in regulating a wide variety of biological programs, including physiologic hypoxic response, oncogenic programs, and translational control. Both HIF-dependent and HIF-independent pathways are highlighted. The review is well structured; however, there are some additional recommended clarifications. Some of these are recommended with the goal of helping non-expert readers; whereas, others illustrate some of the existing inconsistencies in the field.
We thank the reviewer for their comments and we have addressed specific comments below.
Specific Comments:
1) Line 42: The section describing the 3 different isoforms would benefit by clarifying that HIF3a lacks a TAD, and thus either acts as a dominant-negative or transcription-independent regulator of hypoxic response.
As suggested we have clarified this point.
2) Line 49 reads: “The mechanism leading to the activation of HIF was unravelled in 2001 [6,7]”. Although, these two seminal studies did indeed cement the mechanism of hypoxic HIF activation via prolyl hydroxylation, a series of previous findings had begun linking HIF proteolysis to pVHL in hypoxia [for example, Maxwell PH (1999) Nature and Ohh M (2000) Nat Cell Biol.]. Line 49 fails to capture this idea sufficiently.
We have added these references and corrected the sentence to refer to additional work prior to PHD identification.
3) Line 58: “Biochemical characterisation revealed that PHDs have low affinity for molecular oxygen.” This statement needs citations.
We have added a reference to this point.
4) Line 73: “Furthermore, genomic techniques…in response to 2-OGD inhibition.” This statement needs citations to highlight the findings.
We have added several references to this point as suggested.
5) O2 Km values vary significantly between reports, perhaps because of differences in substrate lengths, buffer conditions, etc. As done with Table 1, it might help to put citations in Box 1 to allow readers to identify the experimental conditions under which a given Km was measured.
We thank the reviewer for this point. We have added references and expanded this in a supplementary table.
6) ADO can be added to Box 1 as an oxygen sensor. Masson N etal (2019) Science.
We have added ADO to the table as suggested
7) The role of KDM3A in hypoxia remains confusing. While the authors highlight (line 263) that KDM3A has been shown to support HIF function, and thus is likely to be active in hypoxia; Qian X etal (2019) Mol Cell reported that KDM3A is an oxygen-sensitive histone demethylase. This idea needs to clarified in the text.
We have added a few sentences and reference to address this point as suggested.
8) Links between translational regulation and HIF extend beyond hypoxia. For example, links between HIF and XBP1 [Chen X (2014) Nature] could be highlighted.
We have narrowed the focus of the review to focus more tightly on the role of 2-OGDs, and have therefore removed the section discussing selective translation in hypoxia. This study, while important, and an interesting link between HIF-1α and the UPR through XBP1, is felt to be outside the scope of our discussion.
9) There are a few instances of typographical errors. Some of these include:
- Line 68: extra [G] in, ‘chromatin [G] structure and epigenetics [G] to RNA biology’
- Line 176: extra [G] in, ‘The first discovered mechanism of selective translation was internal ribosomal entry sites (IRES) [G]’
- Line 415: SLC2A1 typo in, ‘Glucose transporter 1 (GLUT1) (gene name SLCA1)’.
We hope we have corrected all these errors now
Reviewer 4 Report
The manuscript by Frost et al. comprises a review of the roles of 2-oxoglutarate dependent dioxygenase enzymes (2-OGDs) as potential oxygen sensors in controlling gene expression in response to hypoxia, via HIF dependent and HIF independent pathways.
This is an important and timely review that would be of great interest to the readers of Cancers.
Major Criticisms
The main criticism I have is that very important statements/assumptions are not always referenced. Notably, Box 1 provides details of the Kms of the proteins, an indication of whether or not an enzyme could act as a potential O2 sensor, and whether said enzymes affect gene expression in hypoxia, with no references, or statements as to how the classifications were deduced. Similarly, the substrates in this table should be better defined (for instance which specific lysine residues on Histone H3 are targeted etc) and also referenced. There is no legend to this box, and it is this unclear why some TETs/JmjCs have not been included.
The section on phenotypes of genetically modified mice lacking these 2-OGDs/disease phenotypes associated with mutations in 2OGDs is extremely useful and interesting. However, some discussion of how the phenotypes might relate to the potential roles of the 2OGDs in the O2-dependent regulation of gene expression was missing (but would seem appropriate). For instance, developmental defects associated with genetic ablation of TET/JmjC enzymes may relate to the critical role of O2 availability in the developing embryo. Similarly, the relevance of the hypoxic microenvironment of solid tumours to aberrant demethylase activity of JmjCs/TETs in cancer should have been better addressed.
Minor criticisms: “Dioxygenases” should be included in the title (to distinguish from other enzymes that might use 2-OG as substrate).
Line 56: it should be made clear that the 2-OGDs require Fe2+. (The ferric ion is inhibitory)
Line 62. The word “Subsequently” is misleading. P4H was identified and characterised long before this.
The term “hypoxia” is relative and is dependent upon cell type. This should be made clearer (throughout) when discussing whether or not the affinity and hence Km of a specific protein for O2 renders it a potential O2 sensor.
The referencing format in Table 1 is not consistent throughout
There are frequent typographical and grammatical mistakes which need to be corrected.
Author Response
The manuscript by Frost et al. comprises a review of the roles of 2-oxoglutarate dependent dioxygenase enzymes (2-OGDs) as potential oxygen sensors in controlling gene expression in response to hypoxia, via HIF dependent and HIF independent pathways.
This is an important and timely review that would be of great interest to the readers of Cancers.
We would like to thank this reviewer for the positive comments, we will address the specific points below.
Major Criticisms
The main criticism I have is that very important statements/assumptions are not always referenced. Notably, Box 1 provides details of the Kms of the proteins, an indication of whether or not an enzyme could act as a potential O2 sensor, and whether said enzymes affect gene expression in hypoxia, with no references, or statements as to how the classifications were deduced. Similarly, the substrates in this table should be better defined (for instance which specific lysine residues on Histone H3 are targeted etc) and also referenced. There is no legend to this box, and it is this unclear why some TETs/JmjCs have not been included.
As suggested, we have added references and the known subtrates for the 2-OGDs where known. We focused on the enzymes that have been characterised in vitro, hence not all 2-OGD being present in Box1. We have also expanded the information into a supplementary Table.
The section on phenotypes of genetically modified mice lacking these 2-OGDs/disease phenotypes associated with mutations in 2OGDs is extremely useful and interesting. However, some discussion of how the phenotypes might relate to the potential roles of the 2OGDs in the O2-dependent regulation of gene expression was missing (but would seem appropriate). For instance, developmental defects associated with genetic ablation of TET/JmjC enzymes may relate to the critical role of O2 availability in the developing embryo. Similarly, the relevance of the hypoxic microenvironment of solid tumours to aberrant demethylase activity of JmjCs/TETs in cancer should have been better addressed.
We thank the reviewer for this comment. Indeed, the phenotypes must result from a complex set of alterations. We have added a sentence in the discussion highlighting the potential of these enzymes also reflecting the metabolism of the cells, being in fact metabolic sensors.
Minor criticisms: “Dioxygenases” should be included in the title (to distinguish from other enzymes that might use 2-OG as substrate).
We have added this to the tittle as suggested.
Line 56: it should be made clear that the 2-OGDs require Fe2+. (The ferric ion is inhibitory)
We have added this information as suggested.
Line 62. The word “Subsequently” is misleading. P4H was identified and characterised long before this.
We have removed this word as suggested.
The term “hypoxia” is relative and is dependent upon cell type. This should be made clearer (throughout) when discussing whether or not the affinity and hence Km of a specific protein for O2 renders it a potential O2 sensor.
Indeed, we have addeed a sentence to the introduction, when we define hypoxia to clarify this point. We thank the reviewer for point this out to us.
The referencing format in Table 1 is not consistent throughout
We have corrected this sotware mistake.
There are frequent typographical and grammatical mistakes which need to be corrected.
We have proofread and corrected the mistakes we could detect.
Round 2
Reviewer 2 Report
The authors have addressed my previous concerns. The article should be accepted upon addressing the following minor changes and the authors should carefully review the manuscript to look for small mistakes (I have picked out some, which are outlined below).
Minor revisions
- On line 98 “possile” should read as “possible”.
- Reference 25 should be included within box 1 in the row for KDM4A under the “O2 KM (µM)” column. Additionally, this study reported the in vitro oxygen sensitivity for KDM4B, which is missing from this table.
- On line 221 “…through histone their…” should read as “… through their histone…”.
- On line 401 “auto-hydroxylate” should read as “auto-hydroxylates”.
- On line 402 “homo-oligomerised” should read as “homo-oligomerisation”.
- On line 430 “K327me1” should read as “K372me1”.
- The enzyme-substrate interactions of KDM3A and KDM3B and the corresponding roles (described within lines 425-433) should also be depicted within Figure 5.
- The authors should adjust the text on line 466 as “JMJD5” is described within the legend but it is not present within the figure. There may also be a few words missing from this sentence.
Author Response
We would like to thank the reviewer for their positive comments. We have addressed the minor points below
- On line 98 “possile” should read as “possible”.
We have corrected this mistake
- Reference 25 should be included within box 1 in the row for KDM4A under the “O2 KM (µM)” column. Additionally, this study reported the in vitro oxygen sensitivity for KDM4B, which is missing from this table.
We have added this information to Box 1
- On line 221 “…through histone their…” should read as “… through their histone…”
We have corrected this mistake, thank you.
- On line 401 “auto-hydroxylate” should read as “auto-hydroxylates”.
We have corrected this mistake
- On line 402 “homo-oligomerised” should read as “homo-oligomerisation”.
We have corrected this mistake
- On line 430 “K327me1” should read as “K372me1”.
We have corrected this mistake
- The enzyme-substrate interactions of KDM3A and KDM3B and the corresponding roles (described within lines 425-433) should also be depicted within Figure 5.
We have added KDM3A to Figure 5 as suggested
- The authors should adjust the text on line 466 as “JMJD5” is described within the legend but it is not present within the figure. There may also be a few words missing from this sentence.
We have corrected this mistake